# Genomic profiling of active vitamin D colonic responses in African- and European-Americans identifies an ancestry-related regulatory variant of *POLB*

David Witonsky[1,º], Bharathi Laxman[2,º], Hina Usman[2], Margaret C. Bielski[2], Kristi M. Lawrence[2], Sonia S. Kupfer[2,3*]

1 Department of Human Genetics, University of Chicago, Chicago, Illinois, United States of America, 2 Section of Gastroenterology, Hepatology & Nutrition, Department of Medicine, University of Chicago, Chicago, Illinois, United States of America, 3 Section of Genetic Medicine, Department of Medicine, University of Chicago, Chicago, Illinois, United States of America

º These authors contributed equally to this work.
* skupfer@medicine.bsd.uchicago.edu

## Abstract

We measured genomic responses to active vitamin D, 1α,25-dihydroxyvitamin D (1,25D), in colonic organoids from individuals of African and European ancestry. Given protective effects of 1,25D for gastrointestinal conditions such as colorectal cancer, organoid cultures enabled evaluation of condition-specific responses in relevant target tissue across individuals of diverse ancestries. We found significant alterations in transcriptional and chromatin accessibility responses to 1,25D treatment, including some with ancestry-associated differences, and also elucidated the role of *cis*-genetic variants on treatment responses. Integration of genomic profiling with genetic mapping found an insertion-deletion variant that explains ancestry-associated differences in 1,25D regulation of *POLB,* an oxidative DNA repair enzyme involved in colorectal carcinogenesis, which also showed signals of positive natural selection. These findings highlight the importance of including diverse individuals in functional genomics studies to identify potential drivers of population-level differences relevant for clinical outcomes, and to uncover functional mechanisms that may be obscured by ancestry variation.

## Author summary

In our study, we aimed to understand how active vitamin D affects colon cells from people with African and European backgrounds. Colon health is important for everyone, especially since conditions like colon cancer can be influenced by vitamin D responses. Leveraging an experimental approach in which we treated colonic organoids, also known as "mini-guts", from people of different

---

**Data availability statement:** The data that support the findings of this study are publicly available from GEO with accession GSE295961.

**Funding:** This study was supported by the National Institute of Health grant R01CA220329-01A1 to SSK. The funder had no role in study design, data collection of analysis, decision to publish or preparation of the manuscript.

**Competing interests:** The authors have declared that no competing interests exist.

backgrounds with active vitamin D, we found that genes and cell structures respond differently to vitamin D, some of which depend on a person's ancestry. We discovered that vitamin D changes the activity of many genes and some of these changes are different depending on a person's ancestry.

We found one specific genetic difference that helps explain why an important gene called polymerase beta, or *POLB*, which is involved in repairing DNA and prevention of colon cancer, responds more strongly to vitamin D in certain populations. Including people with different backgrounds in genetic research is critical to better understand how treatments like vitamin D might work differently for heterogeneous groups and could improve health care for everyone.

## Introduction

Active vitamin D, known as calcitriol or 1α,25-dihydroxyvitamin D (1α,25(OH)$_2$D$_3$ or 1,25D), is a nuclear hormone with effects on a variety of biological processes [1], including protective effects against gastrointestinal (GI) conditions such as colorectal cancer [2–4], inflammatory bowel disease [5], and gut dysbiosis [6]. Circulating levels of the inactive form of vitamin D, 25-hydroxyvitamin D (25(OH)D or 25D), vary substantially across ancestries [7] and may partially account for inter-ethnic variation in the prevalence and severity of multiple vitamin D-linked disorders. However, while serum 25D levels are a proxy for overall vitamin D stores, an exclusive focus on this measure may obscure clinically significant aspects of the active vitamin's impact on human biology. We propose that both individuals and ancestry groups may display substantial variation in the responses of target tissues to 1,25D, irrespective of circulating levels. An individual's genotype and ancestral background could be an important determinant of their response to 1,25D which could influence clinical outcomes and might explain the ambiguous results of vitamin D supplementation trials that fail to control for ancestry and other sources of inherited variation [8–10]. Genomic measures of individual responsiveness to 1,25D, such as differences in chromatin accessibility and transcriptional activity, may therefore be more relevant clinical endpoints than 25D levels [11].

 Prior efforts to characterize genomic responses to vitamin D in humans have mainly been conducted in systems that are relatively easy to sample, such as peripheral blood cells [12,13] and cell lines [14–16]. While these investigations yield meaningful results for certain cell types, they may not recapitulate the context-specific responses in relevant tissues across multiple individuals, limiting their applicability to assess inter-individual responses. Alternative models in relevant tissue types from genetically diverse individuals may enable improved characterization of responses, as we demonstrated in a previous study in primary colon organ cultures [17]. However, those cultures were limited by short viability and mixtures of epithelial and stromal cells. Organoid cultures, derived from colonic epithelial stem cells with longer and more stable viability [18], offer a more robust experimental framework for

studying genomic responses in human colon to environmental factors including 1,25D [19]. Observations of coordinated shifts in chromatin accessibility and transcriptional activity in these cultures following experimental perturbations can be particularly helpful for revealing specific regulatory pathways controlled by treatments of interest such as vitamin D across individuals.

Here, we applied this approach to the question of inter-individual and inter-ancestry variation in 1,25D responses within the colon, comparing treatment responses in colonic organoid cultures from individuals of African and European ancestry. By comprehensively profiling transcriptional and chromatin accessibility responses to 1,25D across heterogeneous individuals, we identified novel context-specific genetic effects that regulate these molecular responses, including ancestry-related regulation of biologically relevant genes for GI health and disease.

## Results

### Vitamin D treatment induces widespread genomic responses in colonic organoids

We characterized genomic responses to 1,25D by comparing gene expression and chromatin accessibility data from vitamin D- and control-treated organoids (Fig 1a). Colonic biopsies from 60 healthy patients were used to generate colonic organoids. After 24 hours in differentiation media, we separately treated replicates of each of the 60 organoid lines with either 100nM 1,25D or vehicle control (0.1% ethanol) based on the results of our pilot study where we had determined optimal treatment dose and duration for this system [19]. We prepared organoids for ATAC-seq and RNA-seq after 4 and 6 hours of exposure to each treatment, respectively. For transcriptional responses, after quality controls (see Methods), we retained a total of 53 lines for transcriptome profiling, including 26 African-Americans (AA) and 27 European-Americans (EA). For chromatin accessibility profiling, we applied additional stringent quality controls (see Methods) and retained ATAC-seq data from 25 individuals, including 12 AA and 13 EA donors (S1 Table).

To identify differentially expressed (DE) genes between 1,25D and vehicle control treatments, we used linear mixed models implemented in the *dream* R software package [20], including individual as a random effects variable and age, sex, batch, population, and cell type composition as fixed effects covariates (see Methods *Model 1*). Cell type proportion was measured as the total fraction of early enterocytes and enterocytes deconvolved from the bulk RNA-seq data using a gene signature matrix derived from a single cell RNA-sequencing dataset (S1 Fig). All organoid lines displayed substantial shifts in expression with 1,25D exposure (S2 Fig). Of 13,163 protein-coding genes tested, 8,981 (68.2%) DE genes were identified at a false discovery rate (FDR) <5% (Fig 1b). Of these, 245 (2.7%) showed an absolute log2 fold change (LFC) with magnitude 1.5 or greater. This highly DE subgroup showed greater upregulation than downregulation, with 195 genes showing increased expression and 50 showing diminished expression — an almost 4-fold difference. The top upregulated DE genes included genes that are known vitamin D target genes via VDR binding: *CYP24A1, TRPV6, FGF19, CYP2B6,* and *CD14*. Top downregulated DE genes included *SOX7, PCK1, SYNE3,* and *SP6* (full list in S2 Table). The gene encoding VDR showed a small but significant downregulation with 1,25D treatment (LFC = -0.35; FDR = $6.2 \times 10^{-17}$).

We performed gene set enrichment analysis of transcriptional responses to characterize the types of biological pathways activated or suppressed in response to 1,25D using SetRank [21]. In total, 128 pathways from 3 databases (KEGG, REACTOME, GOBP) are significantly enriched among DE genes based on adjusted p-values (S3 Table). Top enriched pathways include functions related to transport, signaling by G protein coupled receptor as well as negative regulation of cell proliferation and ion transport (Fig 1c). One disease-associated pathway was significantly enriched among DE genes, namely "pathways in cancer" (KEGG hsa05200) (S2 Fig).

Next, we examined differences in chromatin accessibility between treatment and control conditions across 118,806 ATAC peaks detected in at least 3 of the 50 samples from both treatment conditions. We identified differentially accessible (DA) peaks at an FDR < 5% using a linear fixed-effects model implemented in the DiffBind R package [22], including individual as a fixed effect covariate (see Methods *Model 3*). Like gene expression, organoid lines displayed substantial shifts in chromatin accessibility with 1,25D exposure (S2 Fig). Among the 25 lines that met quality controls, we found 4,142 DA

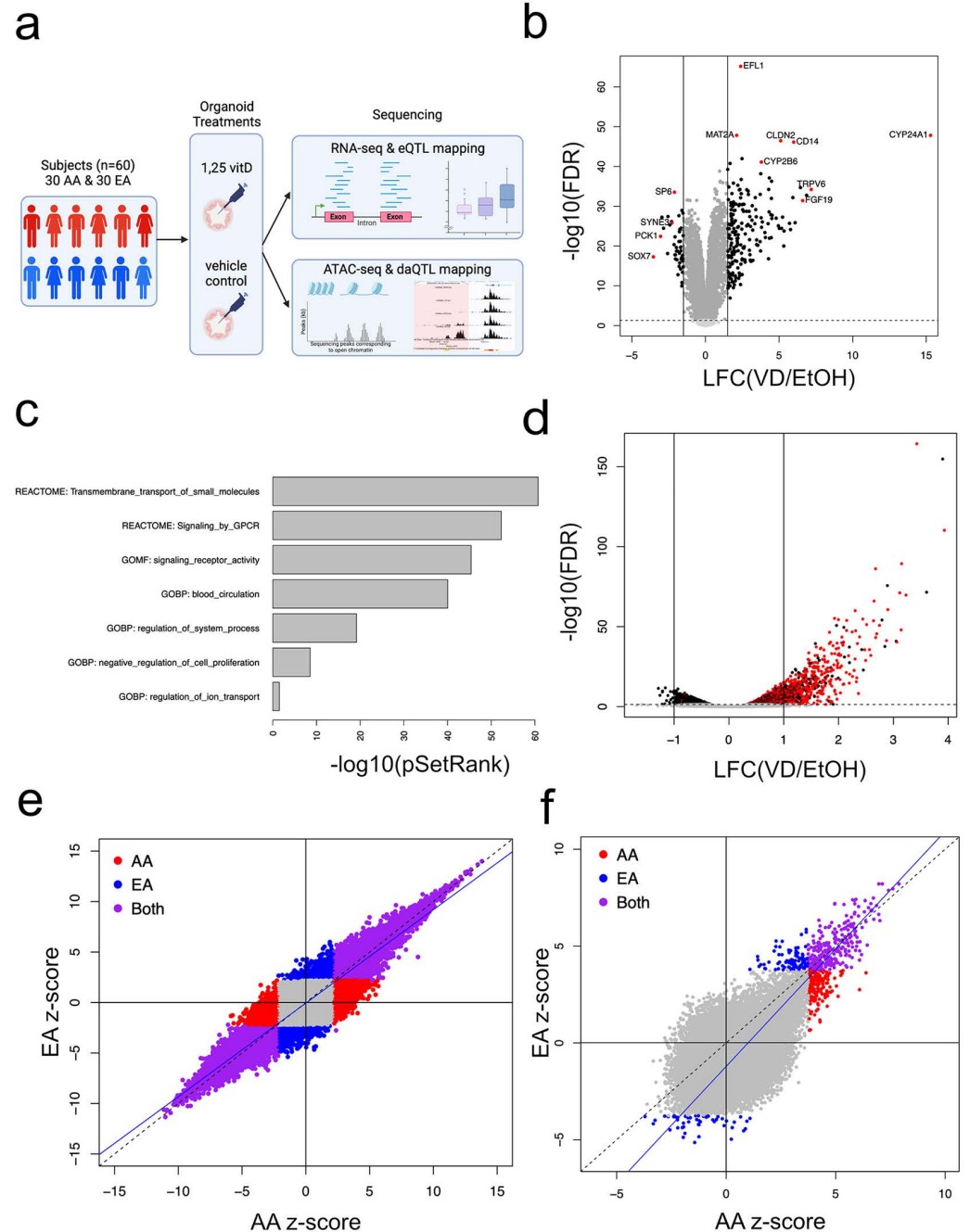

**Fig 1. Genomic responses to 1,25 vitamin D (VD) treatment in colonic organoids. a) Study design.** Colonic biopsies from a total of 60 healthy individuals (30 AA and 30 EA) were obtained during colonoscopy screening exams from which colonic organoids were generated. After 24 hours in differentiation media, we cultured replicates of each organoid line with either 100nM 1,25 vitamin D (VD) or vehicle control (0.1% ethanol). After 4 and 6 hours of exposure to each treatment, we performed ATAC-seq and RNA-seq, respectively. We then mapped QTLs for condition-genotype interaction responses (i.e., reQTL and daQTL). *Created in BioRender. Kupfer, S. (2026)* https://BioRender.com/n5jbbeb. **b) Differentially expressed (DE) genes with VD treatment.** Of 13,163 protein-coding genes tested, 8,981 DE genes were identified at FDR < 5% (black dots). Among genes with larger effect sizes (e.g., |LFC| > 1.5; indicated by horizontal lines), there was an almost 4-fold greater upregulation compared to downregulation. Top upregulated genes included known targets of the VDR such as *CYP24A1, TRPV6, FGF19, CYP2B6,* and *CD14*, while top down-regulated genes included *SOX7, PCK1, SYNE3,* and *SP6* (red dots). **c) Gene set enrichment analysis of transcriptional responses.** A total of 128 pathways from 3 databases (KEGG, GOBP, REACTOME) were significantly enriched among DE genes based on adjusted p-values. Significantly enriched pathways included functions related to

transmembrane transport, signaling by G protein coupled receptors as well as negative regulation of cell proliferation and regulation of ion transport. **d) Differential chromatin accessibility (DA) with VD treatment.** Among 25 organoid lines that met quality controls, a total of 4,142 peaks (3.5%) were DA with 1,25D treatment at FDR<5% (black dots). Of 639 strong DA peaks (e.g., |LFC|>1) (indicated by horizontal lines), 622 (97.5%) showed greater accessibility with 1,25D treatment. The motif for the vitamin D response element (VDRE, defined by a predicted VDR and RXR binding; red dots) was significantly enriched among DA peaks, especially those with positive LFC. **e) Genome-wide VD transcriptional responses by population.** There was significant correlation between z-scores for tested genes with treatment between AA and EA (r=0.94; p<2.2x10^-16). A large number of DE genes were significant in both populations at FDR<5% (purple dots). A smaller number of DE genes were significant in AA (red dots) or EA (blue dots). Line y=x (dashed line). Line of best fit for DE genes: y=-0.06+0.93 (blue line). **f) Genome-wide VD chromatin accessibility by population.** Similar to findings for DE genes, we found strong correlation between z-scores for DA peaks (r=0.86; p<2.2x10^-16). When testing for DA peaks in the populations individually (12 AA lines; 13 EA lines), a total of 582 DA peaks were found in AA or EA at FDR<5%. Of these, 177 DA peaks were significant in AA only (red dots), 141 DA peaks in EA only (blue dots) and 264 DA peaks were significant in both populations (purple dots). Line y=x (dashed line). Line of best fit for DA peaks: y=-1.21+1.21 (blue line). **Abbreviations:** VD, 1,25D vitamin D; EA, European-American; AA, African-American; QTL, quantitative trait loci; DE, differentially expressed; FDR, false discovery rate; LFC, log fold change; DA, differentially accessible.

peaks (3.5%), of which 639 (15.3%) displayed high responsiveness (|LFC|>1). Most of this subset of highly responsive DA peaks (622 peaks, 97.5%) showed greater accessibility in response to 1,25D (Fig 1d and S4 Table). DA peaks showed significant enrichment relative to all peaks in intronic (49.3% vs. 43.8%, respectively; p=2.10x10^-13) and intergenic (41.7% vs. 38.0%, respectively; p=3.32x10^-7) regions. Conversely, DA peaks were significantly depleted in promoter regions (3.6% vs. 12.4%, respectively; p=1.54x10^-89). This depletion was also seen reflected in a broader distribution of distances to the transcription start site (TSS) of the nearest gene for DA peaks relative to peaks that were not DA (S5 Table and S3 Fig). A total of 85 (0.65%) of the tested protein-coding genes were found to have DA peaks falling in their promoters. Of these, 73/85 (86%) were DE, a 1.3-fold enrichment over genes without DA peaks in their promoters (p=8.18x10^-5). We found that the corresponding DE and DA effect sizes for these 73 genes were strongly correlated (r=0.85; p<2.2x10^-16) (S3 Fig), suggesting a probable mechanistic link between 1,25D-induced alterations in gene expression and proximal chromatin accessibility. Roughly the same fraction of DE genes with a DA promoter showed upregulation (48%) as the overall set of DE genes. Despite the observed enrichment of upregulated genes among highly DE genes, this more proximal mediation does not alter the likelihood of the direction of gene regulation.

We further analyzed the sequence content of the complete set of DA peaks to identify regulatory elements potentially controlled by 1,25D using *Homer* [23]. This analysis identified significant enrichment for motifs of 79 transcription factors (TF). The motif for the vitamin D response element (VDRE), defined by the degenerate consensus motif of VDR and its partner RXR, was present in 19.7% of DA peaks compared to only 2.3% of background peaks (p=1.0x10^−481; S6 Table). It should be noted that this analysis used the degenerate consensus motif as defined in *Homer* allowing for variability at multiple positions and a family of VDR-RXR binding sites rather than a single fixed sequence. The VDR-RXR motif was the most significantly enriched among DA peaks relative to background peaks with large effects (i.e., LFC>1; 62% vs 2.8%; p=1.0x10^-423) as well as for peaks with weaker but positive effects (i.e., 0<LFC<1; 30.6% vs 2.5%; p=1.0x10^-313). The VDR-RXR motif showed a non-significant deficit relative to background in DA peaks with reduced effects with 1,25D treatment (i.e., LFC<0; 0.74% vs 3.02%; p=1.0). Mirroring these enrichment patterns, a VDR-RXR motif was found within 250 bp of a DA peak center for 75% of DA peaks with large effects, 46% with moderate effects, and only 3.8% of DA peaks with reduced effects (Fig 3).

In addition to VDRE, 39 TF motifs were enriched with 1,25D treatment for DA peaks with large effects (Table 1). For peaks with opposite effects with 1,25D, the most significantly enriched motif was for KLF5 (52.3% vs 34.1%; p=1.0x10^-66). We note that KLF5 expression is significantly downregulated by 1,25D, which could explain the reduced chromatin accessibility under vitamin D treatment for peaks with the KLF5 motif.

## Inter-ancestry differences in molecular responses to Vitamin D treatment

To test for ancestry-related differences in 1,25D genomic responses, we compared DE and DA results between organoid lines from individuals of African and European ancestry (ancestry proportions shown in S4 Fig). We did not

**Table 1. Transcription Factor Motif Enrichment.**

| Motif Name | Consensus | # of Target Sequences with Motif | % of Target Sequences with Motif | # of Background Sequences with Motif | % of Background Sequences | B-H p-value |
|---|---|---|---|---|---|---|
| VDR,DR3 | ARAGGTCANWGAGTTCANNN | 385 | 62.00% | 3058.5 | 2.80% | 1.00E-423 |
| RARa | TTGAMCTTTG | 436 | 70.20% | 41900.4 | 38.30% | 1.00E-57 |
| COUP-TFII | AGRGGTCA | 314 | 50.60% | 25280.8 | 23.10% | 1.00E-49 |
| EAR2 | NRBCARRGGTCA | 267 | 43.00% | 19504.2 | 17.80% | 1.00E-47 |
| Tbox:Smad | AGGTGHCAGACA | 91 | 14.70% | 2319.6 | 2.10% | 1.00E-46 |
| THRb | TRAGGTCA | 448 | 72.10% | 51844.2 | 47.40% | 1.00E-35 |
| MRE | GGAACAGAVTGTCCT | 212 | 34.10% | 15868.7 | 14.50% | 1.00E-33 |
| COUP-TFII | GKBCARAGGTCA | 264 | 42.50% | 22795.8 | 20.80% | 1.00E-33 |
| ERb,IR3 | RGGTCAGGGTGACCT | 89 | 14.30% | 3717.3 | 3.40% | 1.00E-29 |
| Reverb,DR2 | GTRGGTCASTGGGTCA | 66 | 10.60% | 2124.8 | 1.90% | 1.00E-27 |
| MYRF | AGTGCCTGGCAC | 104 | 16.80% | 6448.6 | 5.90% | 1.00E-20 |
| RORg | WAABTAGGTCAV | 45 | 7.30% | 1393.4 | 1.30% | 1.00E-19 |
| GFY-Staf | RACTACAATTCCCAGAAKGC | 34 | 5.50% | 888.1 | 0.80% | 1.00E-16 |
| YY1 | CAAGATGGCGGC | 31 | 5.00% | 718.5 | 0.70% | 1.00E-16 |
| NFATC2 | WTTTTCCATTGS | 217 | 34.90% | 22751.6 | 20.80% | 1.00E-15 |
| PR | VAGRACAKNCTGTBC | 226 | 36.40% | 24271.7 | 22.20% | 1.00E-15 |
| EBF2 | NABTCCCWDGGGAVH | 116 | 18.70% | 9738.9 | 8.90% | 1.00E-13 |
| CEBP:AP1 | DRTGTTGCAA | 114 | 18.40% | 9964.9 | 9.10% | 1.00E-12 |
| Smad4 | VBSYGTCTGG | 158 | 25.40% | 16452.9 | 15.00% | 1.00E-10 |
| NFkB2-p52 | VGGGRATTYCCC | 78 | 12.60% | 6208 | 5.70% | 1.00E-10 |
| NFkB-p65 | WGGGGATTTCCC | 87 | 14.00% | 7903.7 | 7.20% | 1.00E-08 |
| NFAT | ATTTTCCATT | 98 | 15.80% | 9600.1 | 8.80% | 1.00E-07 |
| Smad2 | CTGTCTGG | 146 | 23.50% | 16531.8 | 15.10% | 1.00E-07 |
| Erra | CAAAGGTCAG | 245 | 39.50% | 33161 | 30.30% | 1.00E-06 |

Top enriched motifs generated from HOMER. Abbreviation: B-H, Benjamini-Hochberg correction for multiple testing.

observe a large genome-wide difference in the transcriptional responses to 1,25D between AA and EA individuals, as shown by strong correlations both between z-scores (r = 0.94; p < 2.2 x 10$^{-16}$) (Fig 1e) and effect sizes (r = 0.97; p < 2.2 x 10$^{-16}$) (S4 Fig) across tested genes. Furthermore, neither population showed a stronger overall absolute response to treatment, as demonstrated by a non-significant between population transcriptional effect size magnitude difference ($|LFC_{AA}| - |LFC_{EA}|$), when compared to the expected null distribution generated from permuted samples (p = 0.28) (S4 Fig).

To identify individual genes with ancestry-related differences in 1,25D treatment responses, we applied a mixed model similar to that used to identify DE genes but with an added treatment by ancestry interaction term. This model yielded highly concordant results regardless of whether ancestry was approximated by a dichotomous population variable or a continuous variable of African ancestry proportion (see Methods *Models 2a & b*), as demonstrated by a strong correlation between interaction term z-scores (S4 Fig). We used a nested approach that focused on the subset of genes that had shown both significant differential expression by 1,25D treatment and by ancestry (n = 81 genes). A total of 8/81 (9.9%) genes showed significant differences in treatment response between AA and EA (Table 2), with four genes (*AMACR*, *TMEM179B*, *SRMS* and *AKAP5*) DE only in AA, one gene (*HS6ST1*) DE only in EA, and three genes (*EPB41L1*, *KYNU*, and *POLB*) DE in both populations, all having 1,25D AA/EA responses in the same direction but with significantly different

PLOS Genetics

**Table 2. Ancestry-related differences in 1,25 vitamin D (VD) treatment responses.**

| Gene Name | LFC$_{AA}$ | LFC$_{EA}$ | LFC$_{AA}$-LFC$_{EA}$ | p-value | adjusted p-value |
|-----------|------------|------------|------------------------|---------|------------------|
| EPB41L1 | -0.456* | -0.283* | -0.173 | 5.21E-05 | 4.22E-03 |
| AMACR | -0.470* | -0.086 | -0.384 | 1.21E-04 | 4.89E-03 |
| KYNU | 3.555* | 1.880* | 1.675 | 5.18E-04 | 1.40E-02 |
| HS6ST1 | -0.053 | -0.411* | 0.358 | 2.19E-03 | 3.72E-02 |
| POLB | 1.870* | 0.988* | 0.882 | 2.30E-03 | 3.72E-02 |
| TMEM179B | 0.133* | 0.001 | 0.131 | 4.05E-03 | 4.72E-02 |
| SRMS | -1.335* | -0.356 | -0.979 | 4.61E-03 | 4.72E-02 |
| AKAP5 | -0.191* | 0.015 | -0.206 | 4.66E-03 | 4.72E-02 |

Using genes that differed by treatment and by ancestry (n = 81 genes), a mixed model approach was applied to identify DE genes with significant interactions between treatment and population, controlling for individual, treatment, ancestry, age, sex, batch and cell composition. Abbreviations: LFC, log fold change (VD/EtOH); AA, African Americans; EA, European Americans.

*LFC adjusted p-value <0.05.

effect sizes. The gene encoding VDR showed no significant difference in differential expression between populations (LFC$_{AA}$-LFC$_{EA}$ = -0.038; FDR = 0.832).

In assessment of ancestry-related differences in chromatin accessibility, we noted results like those for transcriptional responses, with a strong correlation between the population z-scores across DA peaks (Fig 1f; r = 0.86; p < 2.2 x 10$^{-16}$). When testing for DA peaks in the populations individually (n = 12 AA lines; 13 EA lines), a total of 582 DA peaks were found in AA or EA at FDR < 5%. Of these, 177 DA peaks were significant in AA only, 141 DA peaks in EA only and 264 DA peaks were significant in both populations.

## Molecular responses to Vitamin D treatment are genetically regulated

To identify possible genetic contributions to the variation in 1,25D transcriptional responses, we mapped *cis*-expression quantitative trait loci (eQTLs) separately within each treatment condition using the program fastQTL [24]. We then used the Multivariate Adaptive Shrinkage in R package mashr [25] to identify genotype by DE interaction response-eQTLs (reQTLs). At a local false sign rate (lfsr) <0.05 in at least one treatment condition, we identified 41 genes with an reQTL with a > 2x larger posterior mean effect size (pm) in 1,25D, 21 genes with an reQTL with a > 2x larger pm in control, and 230 genes with an reQTL that had significantly nonzero effect sizes in at least one condition but with opposite signs (S7 Table and S5a and S5b Fig).

Of the 13,919 genes tested, 264 (1.9%) had at least one of these three types of reQTL. Among the most significant reQTLs were those for the gene *ZC3HAV1*, a zinc-finger protein previously implicated in colorectal cancer [26] and innate immune responses [27]. The SNP rs12672468, for example, had a pm in control treatment not significantly different from zero (pm = -5.9x10$^{-4}$; lfsr = 0.497), while it had a highly significant pm in 1,25D (pm = -0.594; lfsr = 4.34x10$^{-13}$; S5c and S5d Fig). This strong dependence of genotypic effect size on 1,25D treatment may explain why these reQTLs for *ZC3HAV1* (and others) do not appear as significant eQTLs in sigmoid or transverse colon tissue in the adult Genotype Tissue Expression (GTEx) project database. We also performed overlap analysis with an eQTL mapping study that identified four reQTL in 1,25D treatment in peripheral blood from African Americans and found that rs3848646-*LCCR25* had nominally significant fastQTL p-value in 1,25D (p = 0.011) but not in EtOH (p = 0.228), suggestive of replication.

The significant reQTLs were intersected with the DA regions to determine whether any might exert a direct causal effect on eGene regulation through changes in chromatin accessibility. Six DA regions were found to harbor 7 reQTLs associated with 4 eGenes: *POLB* (8:42190387:A:<CN0>:42194352), *FER1L6* (rs4242349), *SGMS2* (rs28411912; rs28626074; rs28585029), and *ARHGEF28* (rs2112161; rs4703608). All four of these eGenes were found to be significantly

upregulated by 1,25D. The reQTL associated with *POLB* is an approximately 4 kb indel upstream of the *POLB* TSS that encompasses the entire DA peak. To evaluate whether the DA regions mediate the relationship between reQTL genotype and eGene differential expression, we used bmediatR[28], a Bayesian model selection framework for mediation analysis. Of the 7 reQTLs tested, only the *POLB* indel exhibited an appreciable posterior probability (76%) supporting either complete or partial mediation by the DA region (S6a–g Fig).

To determine whether genome-wide chromatin accessibility was influenced by genetic variation, we used matrixEQTL [28] to test for associations between the LFC of each of the 4,142 DA peaks and all imputed variants with a MAF > 0.05 within 100 kb of each peak's summit. We applied both linear and ANOVA models to detect additive and dominant effects, respectively (S8 Table). Significantly associated variants (FDR < 5%) were classified as differential accessibility QTLs (daQTLs). Of the peaks tested, 63 (1.5%) were found to have at least one daQTL under the linear model and 206 (5.0%) under the ANOVA model, with 52 peaks having a daQTL under both models. While approximately half (52.2%) of all DA peaks exhibited reduced chromatin accessibility, we noted that these closing peaks accounted for a significantly higher fraction of daQTL associated peaks: 40/63 (65%; two-tailed hypergeometric test p = 3.6x10$^{-2}$) under the linear model and 146/206 (71%; two-tailed hypergeometric test p = 8.49x10$^{-9}$) under the ANOVA. This skew towards more daQTL-associated closing peaks under 1,25D treatment, may reflect that TFs mediating repressed responses are more sensitive to genetic variation, or that the genomic regions these peaks occupy are less functionally constrained. However, these possibilities require further validation.

To further link DNA sequence variation to the underlying regulatory mechanism contributing to variation in 1,25D transcriptional response, we looked for variants that were both daQTLs and reQTLs for DE genes in our data. We found 33 unique daQTLs (7 under both linear and ANOVA models and 26 under ANOVA alone) that were also reQTLs. All 33 variants were reQTLs for the gene *POLB*, which was strongly DE in our analysis (LFC = 1.47; FDR = 2.80x10$^{-12}$), and daQTLs for a single DA peak approximately 2.3 kb upstream of *POLB* that also showed strong positive response under 1,25D treatment (LFC = 2.42; FDR = 2.50x10$^{-36}$). This DA peak was the same proximal *POLB* peak identified in the above reQTL/DA peak intersection analysis.

### *POLB* colonic responses to 1,25D treatment

*POLB* was of particular interest because of inter-ethnic differences in 1,25D responses. Specifically, we found that the transcriptional response of *POLB* to 1,25D was significantly greater in AA individuals than in EA individuals (LFC$_{AA}$-LFC$_{EA}$ = 0.882, FDR = 3.72x10$^{-2}$; Fig 2a). Correspondingly, the identified mediating peak exhibited increased chromatin accessibility in AA relative to EA individuals after 1,25D treatment and was found to be DA in the AA subset only (Fig 2b). We reasoned that these differences could be genetically regulated. Of the 33 variants that were daQTLs and *POLB* reQTLs, 10 were found to define a core haplotype where pairwise SNP r$^2$ > 0.90 in both YRI and CEU 1000 Genome Project (1KGP) populations (S7a-b Fig). Notably, one of these SNPs (rs3136717; control lfsr = 0.114, 1,25D lfsr = 1.813x10$^{-11}$) had already been shown to lie on a haplotype with large allele frequency differences between Africans and Europeans [29], and another (rs2272733; control lfsr = 0.119, 1,25D lfsr = 8.035x10$^{-12}$) had previously been found to be significantly associated with colorectal cancer [30]. We selected rs2272733 as a tagging SNP for this haplotype because it was the most significant 1,25D-treated *POLB* reQTL genotyped in our data. Among homozygotes or heterozygotes for the rs2272733 ancestral allele, *POLB* showed significant responses to 1,25D. Compared to individuals with at least one copy of the ancestral allele, homozygotes for the rs2272733 derived allele showed no significant differential *POLB* response with 1,25D treatment (Fig 2c and 2d). A parallel accessibility response by genotype was seen for the associated DA peak (Fig 2e). rs2272733 showed large allele frequency differences between individuals of African vs non-African ancestries (Fig 2f).

Taken together, 1,25D responses of *POLB* were of interest due to: 1) ancestry-related transcriptional response differences, 2) a long-range haplotype tagged by a significant response *cis*-eQTL, rs2272733, with large frequency differences

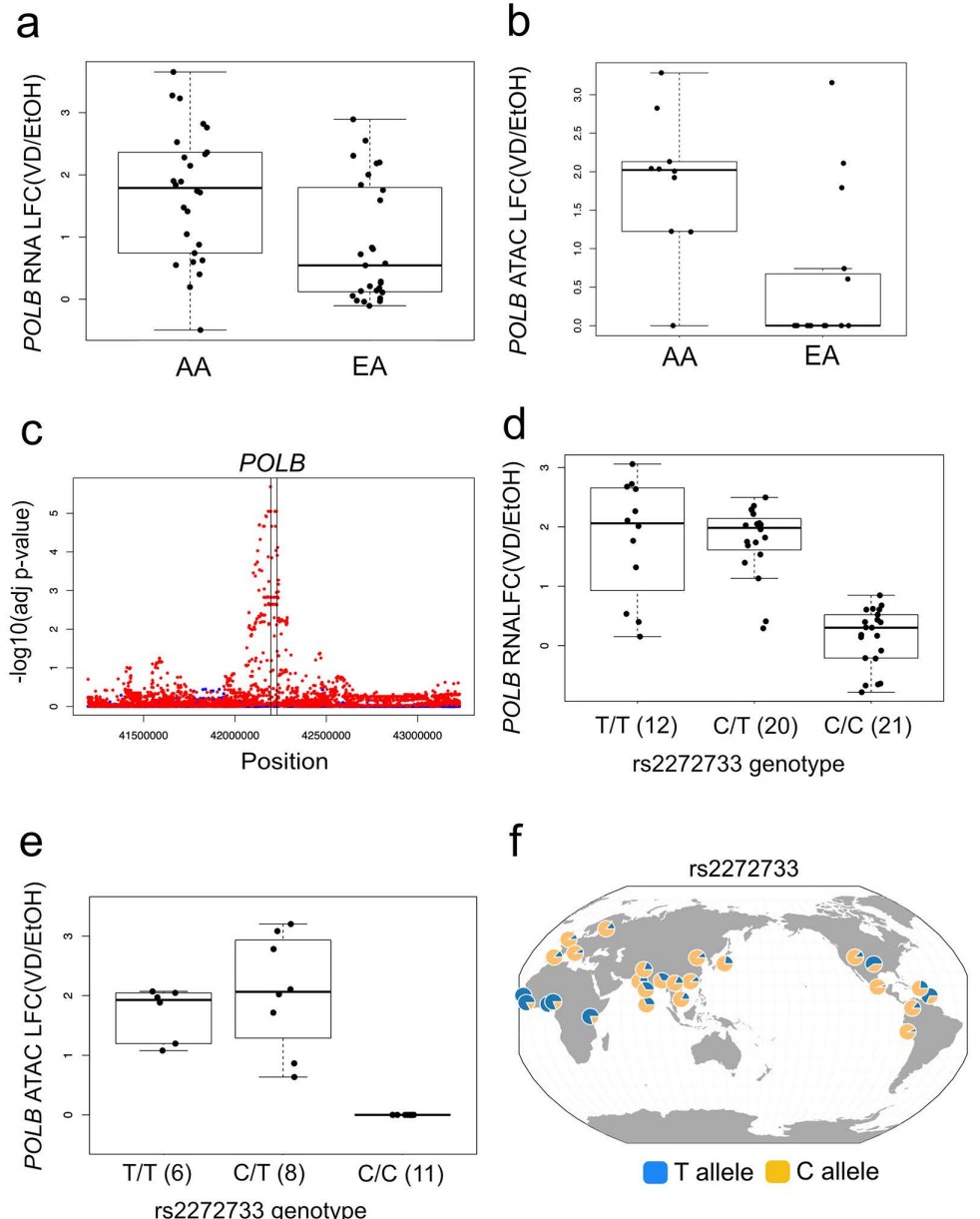

**Fig 2. *POLB* 1,25 vitamin D (VD) responses differed by ancestry and showed a significant vitamin D-specific *cis*-eQTL with large allele frequency differences between African and non-African populations. a) Differential VD transcriptional responses of *POLB* differ by population.** We noted greater VD transcriptional responses of *POLB* in AA compared with EA lines. In AA lines, the *POLB* LFC for VD response was 1.870 (p=3.12x10[-13]), compared to EA lines in which *POLB* LFC was 0.988 (p=7.80x10[-6]). Comparison of *POLB* LFC in AA/EA was 0.88 (p=0.002). **b) Differential VD accessibility peak near *POLB* differs by population.** The most proximal DA peak to the *POLB* gene ("peak A"), lying approximately 2.3 kb upstream of the *POLB* TSS, was noted to have greater accessibility in response to 1,25D in AA (p=2.6x10[-7]) compared with EA lines (p=3.8x10[-1]). The difference between the two populations was significant (t-test p=3.2x10[-3]). **c) *POLB* vitamin D-specific response *cis*-eQTL.** MatrixEQTL adjusted p-values were plotted as a function of physical position for VD (red dots) or control (blue dots) treatment conditions. Vertical lines show the position of the *POLB* gene. **d) rs2272733 as a VD-specific reQTL for *POLB*.** From our daQTL mapping, we found that only the *POLB* proximal peak had significant daQTLs that were also significant reQTLs for a protein-coding DE gene, namely *POLB* . One of these daQTLs is the significant vitamin D-specific response reQTL rs2272733, a SNP genotyped in our dataset, for *POLB* . *POLB* shows differential responses to VD by rs2272733 genotype. The LFC for T/T and T/C genotypes were significantly increased with 1,25D treatment, while the LFC for the C/C genotype was not significantly different from 0 (mean LFC = 0.141, p = 0.21). **e) rs2272733 as a VD-specific daQTL for ATAC peak located proximal to *POLB*.** The ATAC peak shows differential chromatin accessibility to 1,25D by rs2272733 genotype. There were 0 peak reads under both treatments for the C/C genotype. **f) *POLB***

**reQTL, rs2272733, shows large allele frequency differences in African vs. non-African populations.** The significant *POLB* reQTL rs2272733 was notable for large allele frequency differences in African vs. non-African populations. Depicted in this panel are allele frequencies in global populations (T allele = blue; C allele = yellow). In African populations, the average T allele frequency is 77.3%, while the C allele frequency is 22.7%. In European populations, the average frequencies for T and C alleles are 12.4% and 87.6%, respectively. Visualized using the Geography of Genetic Variants Browser. The base layer used in the GG prepared file was from the topojson library. The specific license from the data source: https://github.com/topojson/world-atlas?tab=ISC-1-ov-file#readme. **Abbreviations:** LFC, log fold change; EA, European-American; AA, African-American; p, p-value; adj p-val, adjusted p-value; Pos, position; VD, 1,25D vitamin D; EtOH, ethanol vehicle control.

between populations, 3) a proximal 4 kb indel harboring a DA region that likely mediates between genotype and *POLB* differential expression, 4) significant non-zero 1,25D expression and accessibility responses only in individuals with the ancestral tagging allele of rs2272733, 5) the gene's role as a key polymerase in base excision repair [31] implicated in tumorigenesis [32–34], and 6) association of rs2272733 with colorectal cancer [30]. These findings suggest that an ancestral alteration in *POLB* regulation might be an important driver of inter-ethnic differences in responsiveness to 1,25D, with potentially significant consequences for disease risk.

### Indel variant upstream of *POLB* with large population frequency differences harbors putative regulatory elements and shows signals of natural selection

The core haplotype identified above spans the genes *POLB* and *IKBKB*; however, we found no association with *IKBKB* expression among these SNPs, nor did *IKBKB* show a differential response to vitamin D treatment like that seen in *POLB* (S7c Fig). The 4 kb indel harboring the DA peak ("peak A") is located between the *IKBKB* 3' UTR and the *POLB* TSS (Fig 3a), supporting the inference that rs2272733 is a marker for an upstream *POLB* regulatory element. In African and European populations from the 1KGP (i.e., YRI and CEU; see Methods), this indel was found to be in perfect LD ($r^2 = 1$) with rs2272733 (Fig 3a).

In addition to peak A, a second DA peak ("peak B"; LFC = 1.56; FDR = $5.24 \times 10^{-7}$) fell within the indel region, 3.1 kb upstream of the *POLB* TSS (Fig 3b), signaling a second possible location for our posited vitamin D-responsive regulatory element. Peak A, the stronger, more proximal of the two peaks, harbors a high-scoring predicted VDRE (JASPAR score 610) within 250 bp of the peak's center, making it a strong candidate location for the regulatory element we identified. Peak B, the more distal DA peak, is located in an ENCODE candidate cis-regulatory element with predicted binding of KLF9 and PRDM15. Although neither of these two TF motifs is found to be enriched in DA peaks with positive effect (i.e., LFC > 0), *PRDM15* expression is significantly upregulated by 1,25D, which could provide a trans-acting regulatory mechanism for this secondary site. There was also a second strong predicted VDRE (JASPAR score 564) located in the insertion 3.6 kb upstream of the *POLB* TSS, although it was not within a peak of chromatin accessibility.

Next, we examined the evolutionary features of the indel to characterize its distribution across ancestral populations and assess whether it has undergone selection based on *POLB* regulation. Mirroring the allele frequency difference of the rs2272733 reQTL between YRI and CEU populations, in our sample the insertion was present at high frequency in individuals of African ancestry (63%), but less common in those of European ancestry (20%). The insertion is highly conserved among primates, but not among non-primates, and is flanked by short interspersed nuclear elements (SINEs) (Fig 3c). To explore the putative age of the indel, we analyzed the Allen Ancient DNA Resource (AADR v54.1) database [35] and noted that the insertion was present in all Neanderthal and Denisovan samples — which is expected given that it is ancestral — but absent in the vast majority of non-African samples older than 5,000 years (S9 Table). These results suggest that the deletion is very old, consistent with the allele having arisen in Africa but having already reached high frequency in Europe more than 15,000 before present (BP).

Given the large allele frequency differences of rs2272733 (and the indel variant in perfect LD), we looked for evidence of differential selection pressure between European and African ancestral regions using data from the 1KGP. We noted significant test statistics for CEU-YRI $F_{ST}$ (0.586; empirical p = $3.0 \times 10^{-3}$), CEU-YRI cross-population

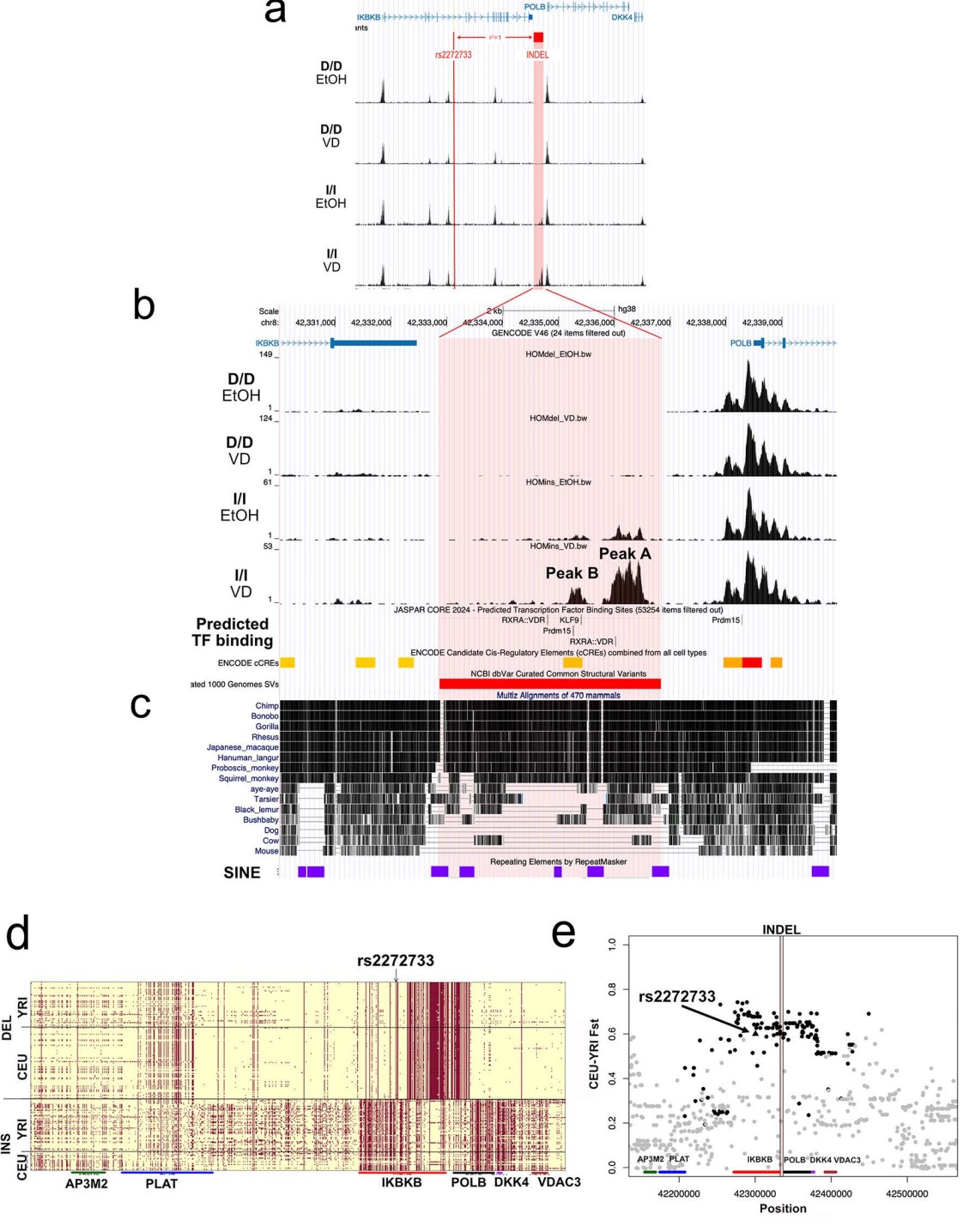

**Fig 3. Indel upstream of *POLB* with large population frequency differences harbors putative regulatory elements and shows signals of natural selection. a) rs2272733 is in linkage disequilibrium with an indel.** The SNP rs2272733 is located in an intron of the gene *IKBKB*, while the associated *POLB* proximal peak ("peak A") is located in a region that overlaps a 4 kb polymorphic 1KGP structural indel variant located between the *IKBKB* 3-prime UTR and the *POLB* TSS. The indel was in perfect LD ($r^2 = 1$) with rs2272733 in CEU and YRI. **b) Insertion harbors two 1,25 vitamin D (VD) responsive peaks with predicted VDREs.** In the insertion, a second, weaker DA peak located 3.1 kb upstream of the *POLB* TSS was identified ("peak B"). Peak B is located in an ENCODE candidate cis-regulatory element with predicted binding of Klf9 and Prdm15. A second strong predicted VDRE is

also located in the insertion 3.6 kb upstream of the *POLB* TSS but is not within a peak of chromatin accessibility. **c) Insertion found to be highly conserved among primates.** The insertion was found to be highly conserved among primates, but not among non-primates. The insertion was flanked by short interspersed nuclear elements (SINEs; shown in purple boxes). **d) Indel polymorphism individual haplotypes by ancestry.** Comparison of individual haplotypes by ancestry showed haplotype structure more consistent with selection of the derived allele (deletion) outside of Africa. **e) SNPs with highest $F_{ST}$ associated only with *POLB* responses.** Looking at the highest $F_{ST}$ SNPs in the 200 kb region surrounding the *POLB* indel, we find that they are in high or perfect LD with the indel, significantly associated with *POLB* (black line) expression in 1,25D treatment, and not associated with the expression of any of the nearby genes *AP3M2* (green line), *PLAT* (blue line), *IKBKB* (red line), *DKK4* (purple line), and *VDAC3* (brown line). Combined, these observations suggest that any selection on the indel is likely due to its influence on *POLB* expression. Dot color same as associated eGene color; gray dots are not significant eQTLs for any of the shown genes. Triangle represents rs2272733. **Abbreviations:** 1KGP, 1000 Genome Project; D/D, homozygous deletion; I/I, homozygous insertion; VD, 1,25D vitamin D; EtOH, ethanol vehicle control; SINE, short interspersed nuclear elements; DEL, deletion; INS, insertion; $F_{ST}$, fixation index; VDRE, vitamin D response element.

extended haplotype homozygosity (XP-EHH) (2.548; empirical $p = 1.8 \times 10^{-2}$), CEU Tajima's D (-1.913; empirical $p = 2.0 \times 10^{-2}$). Taken together, these results are consistent with a positive selection signal within the CEU population. However, we do not find significant statistics for CEU iHS (-0.806; empirical $p = 2.1 \times 10^{-1}$), YRI iHS (0.273; empirical $p = 3.9 \times 10^{-1}$), or YRI Tajima's D (-0.932; empirical $p = 2.0 \times 10^{-1}$). Comparison of individual haplotypes by ancestry showed haplotype structure more consistent with selection of the deletion outside of Africa (Fig 3d), and though neither iHS signal is significant, a negative value in CEU and a positive value in YRI is consistent with selection on the derived allele outside of Africa.

Looking at the highest $F_{ST}$ SNPs in the 200 kb region surrounding the *POLB* indel, we find that they (1) are in high or perfect LD with the indel, (2) are significantly associated with *POLB* expression in 1,25D treatment, and (3) are not associated with the expression of any of the nearby genes *AP3M2*, *PLAT*, *IKBKB*, *DKK4*, and *VDAC3* (Fig 3e). Combined, these observations suggest that any selection on the indel is likely due to its influence on *POLB* expression. While the tagging SNP, rs2272733, was not a significant *cis*-eQTL for any gene in colon tissues (sigmoid and transverse colon) in GTEx, consistent with our finding of this variant as an interaction reQTL, we noted that rs2272733 was a significant *cis*-eQTL for *POLB* in several GTEx tissues (e.g., skin, esophagus, testis, brain, adipose tissue, lung, nerve, skeletal muscle) (S8a Fig). Effects on *POLB* expression in skin, esophagus and, to a lesser extent, testis were in the same direction as effects found in this study (S8b Fig), but opposite in direction in other tissues (S8c Fig). While rs2272733 associations were strongest for *POLB,* other genes, including *RPL5P23, PLAT* and *IKBKB,* were also associated in different tissues in GTEx. Additional studies are needed to understand the role of vitamin D in responses of *POLB* across different tissues to elucidate the observed signals of selection.

## Genotype-specific *POLB* vitamin D responses and VDR binding in indel region

To confirm that the *POLB* responses to 1,25D correlated with indel genotype, we identified organoid lines with homozygous deletion (D/D), heterozygotes (I/D) and homozygous insertion (I/I) based on high-quality imputation results and inferred by rs2272733 genotype which is in perfect LD ($r^2 = 1$) with the indel. We performed quantitative PCR (qPCR) to measure *POLB* expression at 6 and 24 hours after 1,25D treatment in D/D (n = 5), I/D (n = 6) and I/I (n = 7). We also used Western blot to measure POLB protein levels in D/D (n = 5), I/D (n = 5) and I/I (n = 7). We observed mRNA induction of *POLB* by 1,25D at both 6 hours and 24 hours (Fig 4a and 4b, respectively). There were significant differences in levels of *POLB* expression between all genotypes. No differences in responses by genotype were noted between 6 and 24 hours (Fig 4c).

*POLB* protein expression was not upregulated by 1,25D at 6 hours in any genotype (Fig 4d and 4e). At 24 hours, we noted differential *POLB* upregulation by genotype, with significantly enhanced expression in the treatment group among I/I lines compared to D/D lines and intermediate expression in the I/D lines (Fig 4d and 4f). Only the I/I lines showed significant differences in protein expression between 6 and 24 hours (Fig 4g).

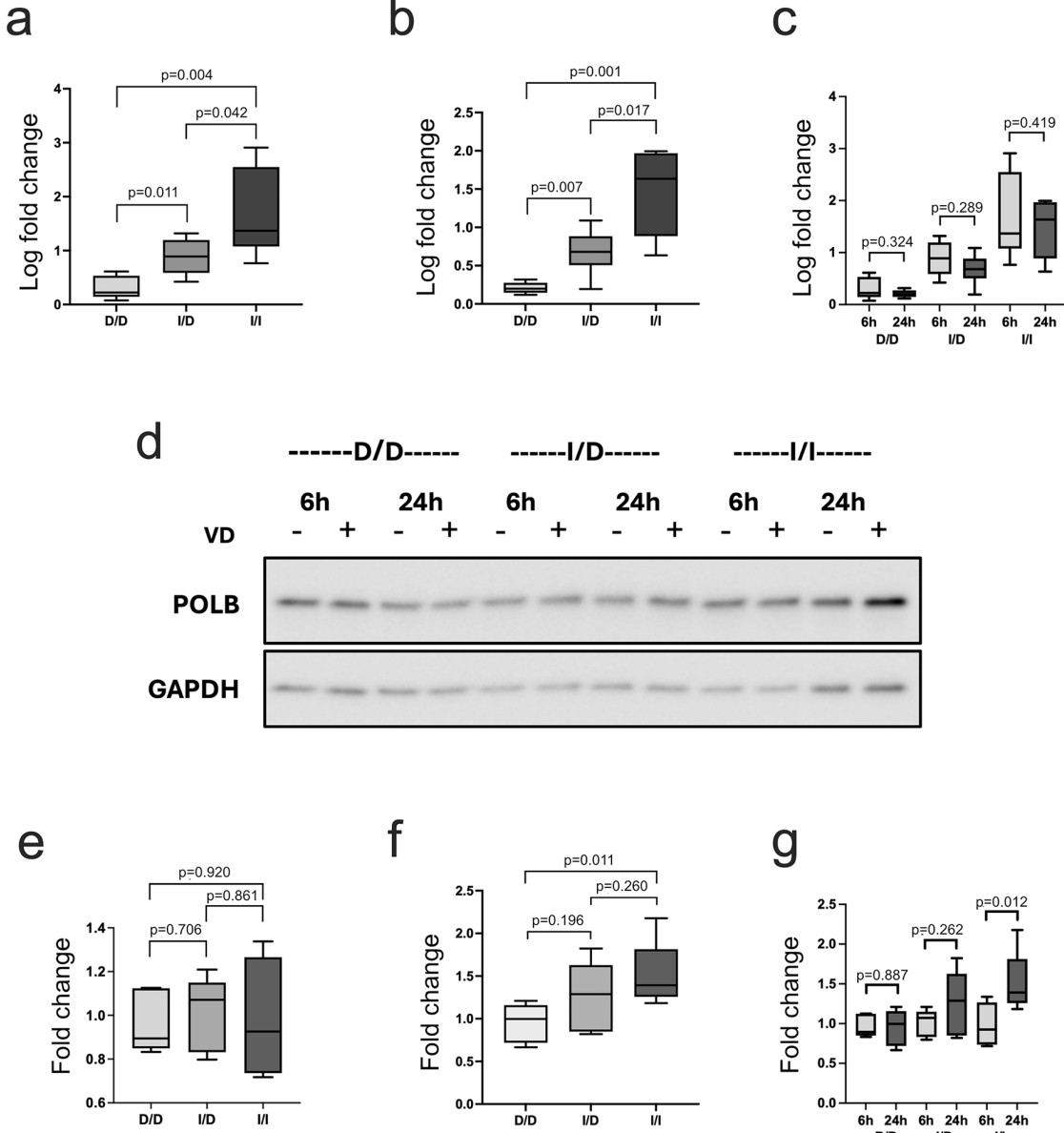

**Fig 4. Genotype-specific *POLB* vitamin D responses and VDR binding.** For these analyses, the indel genotype was defined by high-quality imputed genotype data (posterior probability for imputed genotype >0.999). Imputed genotypes matched genotype of rs2272733, a SNP in perfect linkage disequilibrium ($r^2 = 1$) with the indel in 1KGP CEU and YRI. **a-c) 1,25 vitamin D (VD) responses of *POLB* in organoids at 6 and 24 hours by genotype.** In order to measure *POLB* mRNA expression in response to VD by indel genotype at 2 time points, we treated 5 organoid lines representing homozygous deletion (D/D), 6 heterozygotes (I/D) and 7 homozygous insertion (I/I) for 6 (4a) and 24 (4b) hours and measured *POLB* expression by qPCR. Box plots show log fold change expression of *POLB* stimulated by VD in indel genotypes. The levels of expression differed significantly between the 3 groups at the 2 time points tested with the highest, intermediate and lowest seen in I/I I/D and D/D genotypes respectively. No significant changes in *POLB* expression were noted between the time points for all the genotypes (4c). **d) Western blot of POLB protein expression by indel genotype at 6 and 24 hours.** To measure POLB protein expression in response to VD by indel genotype (defined by rs2272733 genotype) at 2 time points, we treated D/D, I/D and I/I organoids for 6 and 24 hours and measured POLB protein level by Western blotting. Shown are the representative images of blot containing the 3 treated indel genotypes for the 2 time points, probed with antibodies to POLB and GAPDH. Highest level of POLB protein was seen in I/I genotype at 24 hours. **e-g) VD responses of POLB protein in organoids at 6 and 24 hours by indel genotype.** For the 6-hour time point, we treated 5 D/D, 5 I/D and 6 I/I organoid lines. For the 24-hour time point, we treated 5 D/D, 5 I/D and 7 I/I organoid lines. **4e)** Fold change expression of POLB protein by indel genotype for 6 hours. **4f)** Fold change expression of POLB protein by indel genotype for 24 hours. **4g)** Organoids at 6 hours showed no differences in POLB protein levels by indel genotype. The level of POLB protein was significantly higher in the I/I genotype only at 24 hours. **Abbreviations:** 1KGP, 1000 Genome Project; D/D, homozygous deletion; I/D, heterozygote; I/I, homozygous insertion; VD, 1,25D vitamin D.

To identify the specific elements within the *POLB* insertion that are responsible for the observed genotype-specific 1,25D response, we dissected the transcriptional activity of several regions of interest identified by our ATAC-seq profiling of differential chromatin accessibility. To do this, several vector constructs were designed that included: 1) peak A, the proximal DA peak harboring a predicted VDRE, 2) peak B, the distal DA peak without a predicted VDRE, 3) a second predicted VDRE not in a DA peak, and 4) the region encompassing both peak A and B as well as the two predicted VDREs (Fig 5a). We transfected these constructs into 293FT cells, which we exposed to 1,25D or control. After 24 hours, increased transcriptional activity for constructs 1, 3 and 4 was noted, while construct 2 did not show significant transcriptional activity compared to vector control (Fig 5b). Notably, the construct that included both chromatin peaks and predicted VDREs showed increased activity compared to the constructs containing each of these elements individually, suggestive of an additive effect of these regions of interest within the insertion.

To provide additional evidence supporting differential VDR binding by indel genotype, chromatin immunoprecipitation (ChIP) assays were performed using organoid lines with homozygous deletion, heterozygous and homozygous insertion. Organoids were treated with 1,25D or vehicle control, subjected to ChIP assay and PCR using primers designed to amplify the sequence encompassing the predicted VDRE in peak A. When VDR antibody was added with 1,25D treatment, there was evidence of binding noted in the heterozygote and homozygous insertion lines, while no binding was noted in the homozygous deletion line (Fig 5c). As a positive control, VDR binding was assessed using a proximal human *CYP24A1* promoter region (-252 to -51 bp) previously confirmed as a VDR binding site by ChIP-PCR in a colorectal cancer cell line [36]. This region showed evidence of VDR binding with 1,25D treatment across all genotypes. As a further control, both sequences were amplified from input DNA of all the organoid lines except for the sequence encompassing predicted VDRE in peak A from the homozygous deletion line as expected.

## Discussion

Circulating levels of 25D, the inactive form of vitamin D, are known to vary between individuals of diverse ancestries influenced by genetic and environmental factors [37]. In contrast, much less is understood about inter-individual variation in responses to the active form of vitamin D, 1,25D, that mediates a number of important biological functions including protective effects against GI malignant and inflammatory conditions [2–6]. While serum 25D levels are a proxy for vitamin D stores, focus on this measure alone could obscure clinically significant aspects of the 1,25D's impact on human biology [11]. Our genomic evaluation of 1,25D treatment responses in colonic organoids from individuals of African and European ancestry found a large number of significant alterations in gene expression and chromatin accessibility. We also characterized the role of *cis*-genetic variants on these treatment responses. Integration of results from genomic responses and QTL mapping identified an indel that explains ancestry-associated differences in the vitamin D regulation of *POLB* with signals of positive natural selection. These findings underscore the importance of including samples from genetically diverse individuals in functional genomic studies [38] in order to identify potential drivers of population differences that could be relevant for clinical outcomes and to identify new functionally significant mechanisms that might be obscured by solely focusing on individuals from a single ancestral population.

Our results confirm that short-term 1,25D treatment has broad genomic effects in the colonic epithelium [39], some of which are genetically regulated. Transcriptional responses to 1,25D were primarily in the direction of gene upregulation and increased chromatin accessibility for the most highly DE genes and DA peaks, respectively. DA peaks were enriched in intronic and intergenic regions relative to promoter regions which supports previous reports of vitamin D target gene regulation via super enhancer regions [40]. VDR appeared to be the dominant mediator of genomic effects, and, while several other TFs were found to be enriched among DA peaks, TFs previously reported to co-localize with VDR in a leukemia cell line (i.e., PU.1, CEBPA, GABPA) [41] were not found to be enriched in the colon. Eight candidate genes including *POLB* showed significant differences in vitamin D responses by ancestry; though, results showed that genomic 1,25D responses were broadly comparable across individuals of African and European

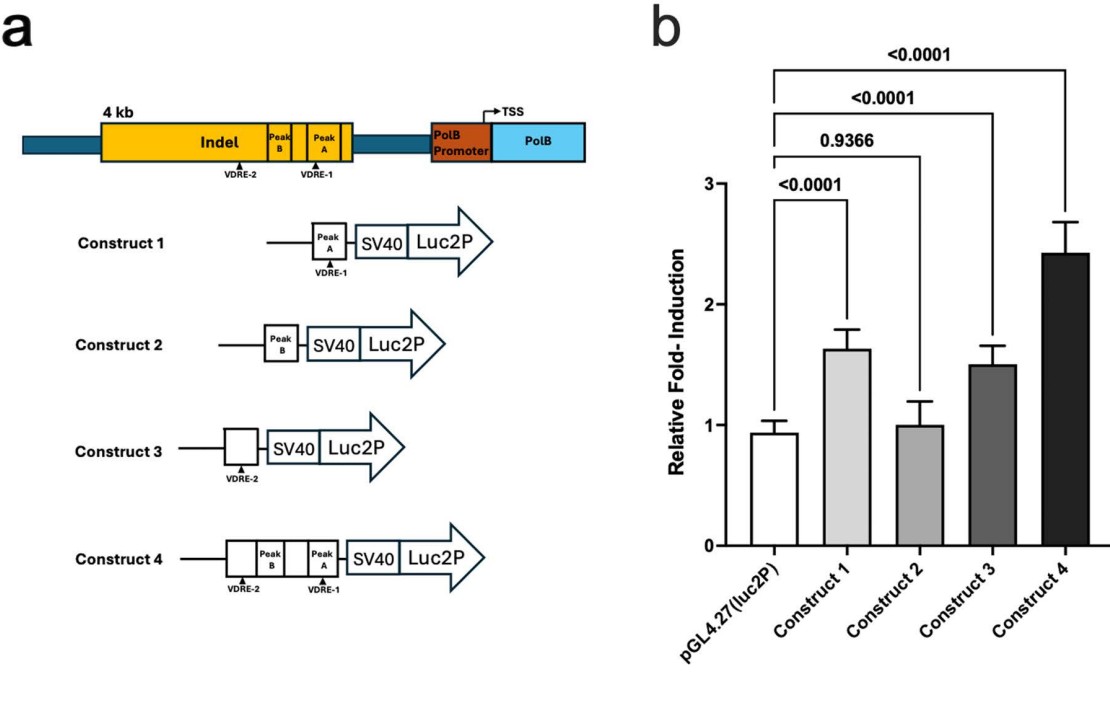

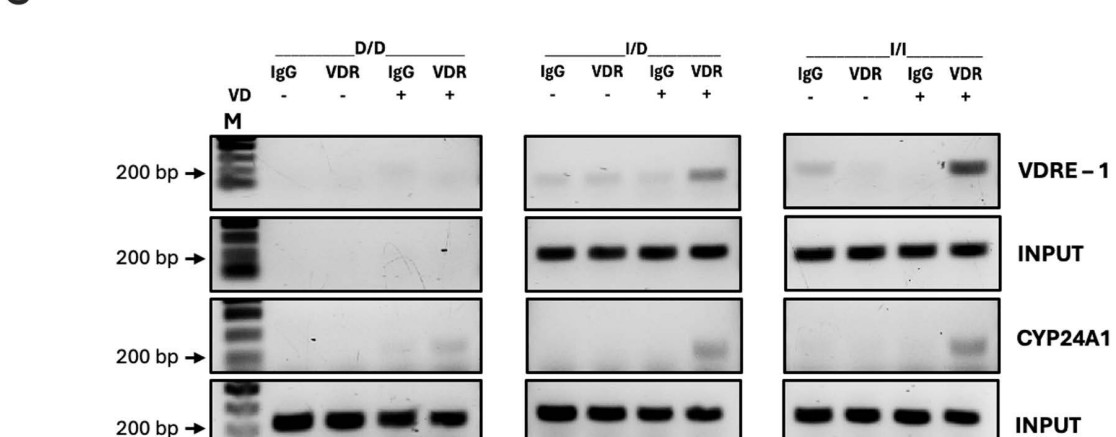

**Fig 5. 1,25 vitamin D (VD) treatment shows differential transcriptional activity and VDR binding by indel genotype.** For these analyses, the indel genotype was defined by high-quality imputed genotype data (posterior probability for imputed genotype >0.999). Imputed genotypes matched genotype of rs2272733, a SNP in perfect linkage disequilibrium ($r^2 = 1$) with the indel in 1KGP CEU and YRI. **a) Indel region luciferase assay constructs.** Based on results from the ATAC-seq profiling showing differential chromatin accessibility with 1,25D, transcriptional activity of specific regions of interest within the *POLB* insertion were dissected. To do this, several vector constructs were designed that included: 1) the DA chromatin peak harboring a predicted VDRE ("peak A"), 2) the second DA chromatin peak without a predicted VDRE ("peak B"), 3) a second predicted VDRE not in a DA peak, and 4) the region encompassing both peaks A and B as well as the two VDREs. **b) *POLB* transcriptional activity for different indel constructs.** After 24 hours of VD treatment, increased transcriptional activity for constructs 1, 3 and 4 were noted, while construct 2 did not show significant transcriptional activity compared to vector control. Notably, the construct that included both chromatin peaks and predicted VDREs showed increased activity compared to each construct individually, suggestive of an additive effect of these regions of interest within the insertion. **c) VDR binding in the indel region by chromatin immunoprecipitation (ChIP) assay.** To provide additional evidence to support differential VDR binding by indel genotype, ChIP assays were performed using organoid lines with homozygous deletion, heterozygous and homozygous insertion. Organoids were treated with or without VD, subjected to ChIP assay, and PCR performed with purified immunoprecipitated chromatin DNA using primers designed to amplify the sequence encompassing the predicted VDRE in peak **A.** When VDR antibody was added with VD treatment, there was evidence of binding noted in the I/D and I/I lines, while no

binding was noted in the D/D line. As a positive control, VDR binding was assessed for a *CYP24A1* promoter region binding site that showed evidence of VDR binding with 1,25D treatment across all genotypes. As a further control, both sequences were amplified from input DNA of all the organoid lines except for the sequence encompassing the predicted VDRE in peak A from the D/D line as expected. **Abbreviations:** 1KGP, 1000 Genome Project; D/D, homozygous deletion; I/D, heterozygote; I/I, homozygous insertion; VD, 1,25D vitamin D; VDRE, vitamin D response element; VDR, vitamin D receptor.

ancestry. Our results contrast with large ancestry-specific differences in responses to glucocorticoids [42] and infections [43–45] across individuals of African and European ancestry in blood and immune cells, which might reflect differences based on selective pressures.

While genetic regulation of circulating levels of 25D has been documented [37], our finding of significant interaction reQTLs across individuals underscores the role of genetic variation in regulation of 1,25D responses, irrespective of 25D levels. For example, the SNP rs12672468 was found to be a vitamin D-only response eQTL associated with expression of *ZC3HAV1* encoding a zinc-finger protein previously implicated in colorectal cancer [26] and innate immune responses [27] that highlights potential new mechanisms of 1,25D actions in the colon that could contribute to differences in disease risks. The only other eQTL mapping study of 1,25D treatment that included African Americans was performed using peripheral blood cells and, of the four interaction reQTLs in this study [12], we show evidence of replication for one (25%) of these reQTLs despite differences in experimental approaches and tissue types.

Integration of results from genomic profiling and QTL mapping identified a novel indel that likely explains observed ancestry-related differences in 1,25D responses of *POLB,* a gene encoding a key polymerase involved in base excision repair. For individuals without at least one insertion (ancestral) allele, *POLB* expression is largely uncoupled from 1,25D exposure, a significant alteration to *POLB* regulation that may be unique in the evolutionary history of primates. Based on our reporter assay and ChIP results, we hypothesize that the regulatory activity occurs via a VDRE located 2.2kb proximal to the *POLB* promoter within the insertion. A second significant DA peak in the insertion did not have evidence of direct VDR binding but did contain a motif for a different transcription factor, PRDM15, that was significantly upregulated by 1,25D, which suggests the possibility of a trans-acting regulatory mechanism for this second peak. Future work will precisely define genetic mechanisms of 1,25D regulation in this region.

The insertion was found to be highly conserved among primates and flanked by short interspersed nuclear elements (SINEs). It is likely that this novel regulatory region was introduced into primates through SINE retrotransposition, specifically Alu-Alu mediated rearrangements [46]. The deletion appears to be present prior to migration out of Africa but rose relatively quickly to higher frequency outside of Africa due to some unknown greater selective pressure. While the exact selective pressures are not known, our results suggest that regulation of *POLB,* rather than other genes in the region such as *IKBKB*, could have been the target for selection, though this hypothesis requires validation in non-colonic tissues. While biological consequences of *POLB* regulation by 1,25D remain to be determined, our findings represent a potential novel mechanism by which 1,25D exerts protective effects against carcinogenesis and, possibly, also inflammation related to oxidative stress in the human colon.

The identification of a vitamin D-dependent regulatory region of *POLB* was particularly interesting given that *POLB* encodes a key enzyme in base excision repair of oxidative DNA damage that is directly involved in colorectal carcinogenesis [32–34]. Studies in both mice [47] and humans [48] have shown that vitamin D protects against oxidative stress-induced DNA damage in the colon, and *POLB* could represent a novel mechanism underlying this protective role. Moreover, the 1,25D-specific reQTL, rs2272733, a tagging SNP for the indel, was previously found to be significantly associated with colorectal cancer; specifically, a protective effect was reported for the T allele [30] (that tags the insertion). In contrast, in African American head and neck cancer patients, the T allele of rs2272733 was associated with poorer prognosis and treatment responses [49]. We believe that the high $F_{ST}$ and association signals that have been previously reported by others for SNPs in this region are likely driven by their linkage disequilibrium with the indel harboring the *POLB* regulatory element. These contrasting association signals could be explained by pleiotropic tissue- and

context-specific effects of vitamin D regulation of *POLB*. Given the central role of *POLB* in DNA base excision repair, even minor perturbations in gene regulation could have significant biological consequences.

Strengths of this study include a paired treatment design that controls for confounders, focus on 1,25D responses on the human colonic epithelium, multi-level treatment response data on chromatin accessibility and differential gene expression in organoid lines from the same individuals, and inclusion of individuals of different ancestral backgrounds. We acknowledge several limitations including assessment of epithelial responses *in vitro* that might not completely recapitulate responses *in vivo*, lack of data on responses in stromal or immune cells that could have relevance for vitamin D's biological actions in the human colon, and a moderate sample size that could lack sufficient power to detect all treatment responses especially when divided by ancestry. Despite these limitations, the study nonetheless identified inter-individual 1,25D responses including the novel ancestry-related regulatory region for *POLB* responses.

In summary, our work leveraging human colonic organoids provides important new insights into mechanisms of context-specific genetic regulation of 1,25D responses in the colonic epithelium. Our results highlight inter-individual differences in responses to the biologically active form of vitamin D on a tissue-specific level, irrespective of serum levels. If confirmed, these findings could inform future efforts to more precisely predict treatment responses for vitamin D-related conditions such as colorectal cancer. We highlight the importance of including diverse individuals in functional genomics research based on identification and mechanistic characterization of a regulatory indel that explains differences in ancestry-related *POLB* responses, findings that broaden understanding of vitamin D regulatory functions with direct implications for health and disease across diverse individuals.

## Materials & methods

### Ethics statement

The study was approved by the University of Chicago Institutional Review Board, protocol IRB24–0931. All participants provided informed consent and de-identified samples were used in the experiments.

### Study participants

Organoids were derived from rectosigmoid colonic biopsies obtained from consenting participants undergoing screening colonoscopy. Lines from a total of 60 participants were included, comprising equal numbers of women (n = 30) and men (n = 30) as well as self-identified Black/African Americans (n = 30) and non-Hispanic Whites (n = 30). The average age of participants was 54.0 years (standard deviation, SD, 9.4) and 57.2 years (SD 9.9) for female and male participants, respectively. The final samples included in downstream analyses were determined by quality controls described in more detail in the Bioinformatic section.

### Organoid cultures

Organoids were derived from colonic biopsies using a protocol adapted from Sato et al., as previously described [18,19]. Organoids were cultured on Biolite Multidish (Fisher Scientific, IL), embedded in six 30 μL droplets of Matrigel, cultured in 1.5 mL of the organoid media in an incubator at 37°C and 5% $CO_2$. Prior to treatments organoids were incubated in basal media for 24 hours to enable differentiation.

### Treatments

For mRNA expression, organoids were treated with 100 nmol/L 1,25D (Enzo Life Sciences, Farmingdale, NY) or vehicle control (0.1% Ethanol) for 4 hours (for ATAC-seq) and 6 hours (for RNA-seq). For qPCR and Western blotting, organoids were treated for 6 and 24 hours.

## RNA-sequencing

Organoids were harvested using cold Advanced DMEM, pipetted up and down 10 times, moved to centrifuge tubes, and then spun at 400 g for 5 min at 4°C. Upper media and Basement Membrane Extract (BME) were carefully aspirated and discarded. mRNA was then isolated from cells using the RNeasy Plus Mini Kit (Qiagen, Germantown, MD) according to the manufacturer's protocol. RNA quality and quantity were assessed using an Agilent Bio-analyzer with RNA integrity numbers (RIN) of >9 for all samples. RNA-seq libraries were prepared using Illumina mRNA TruSeq Kits as protocolled by Illumina. Library quality and quantity were checked using an Agilent Bio-analyzer, and the pool of libraries was sequenced using an Illumina NovaSeq6000 (paired end 100 bp) using Illumina reagents and protocols at the University of Chicago Genomics Facility.

## ATAC-sequencing

Organoids were harvested at the appropriate time point with collagenase IV and treated with 200 U/mL DNase I (Worthington #LS002007) for 30 minutes. Cells were spun for 5 minutes and DNase 1 supernatant was removed. Cells were then resuspended in cold PBS. The cell density and live cell percentages were measured and then 50,000 live cells per replicate were aliquoted. The cells were then treated using the Omni-ATAC protocol [50] with some modifications. Briefly, the cells were treated with lysis buffer (10 mM Tris-HCI pH 7.4, 10 mM NaCl, 3 mM $MgCl_2$, 0.1% IGE-PAL CA-630) and then suspended in the transposition reaction mix (50% TD Tagment DNA buffer, 5% TDE1 Tagment DNA Enzyme, 33% PBS) for 30 minutes. Libraries were prepared using the Nextera Index kit (Illumina #15055289) and were sequenced on the Hi-Seq 4000 platform (50 bp single end reads) at the University of Chicago Genomics Facility.

## Genotyping

Genomic DNA was extracted from blood samples obtained from 60 participants (3 participants did not have available blood samples) and genotyped on the InfiniumOmniExpress-24v1-3_A1 microarray (Illumina; San Diego, CA) which included 714,238 SNPs. Genotype data were used to ascertain genetic ancestry and determine concordance with self-reported ancestry. To increase the density of sites for the purpose of eQTL mapping, genotype imputation was performed with IMPUTE2 [51] using data from 1KGP [52] haplotypes as a reference. Weighted means (dosages) of IMPUTE2's estimated posterior genotype probabilities were calculated and sites were then filtered by minor allele frequency (>0.05), leaving approximately 8.5 million loci.

## Cell culture

For luciferase assays, 293FT cells were cultured at 37°C in 5% CO2 in DMEM, high glucose (Thermo Fisher Scientific, Waltham, MA) supplemented with 10% FBS and 1% penicillin and streptomycin.

## RNA isolation, reverse transcription and qPCR

After the treatment, RNA was isolated using RNeasy Plus Mini Kit from Qiagen (Hilden, Germany) as per the manufacturer's guidelines. Isolated RNA was reverse transcribed into cDNA using a high-capacity cDNA reverse transcription kit from Applied Biosystems, Foster City, CA. qPCR was performed on QuantStudio 6 Flex Real-Time PCR System (Applied Biosystems, Foster City, CA) using TaqManGene Expression assays (*POLB*: Hs01099715_m1 and *GAPDH*: Hs99999905_m1) from Applied Biosystems, Foster City, CA. Following steps were used for amplification of the target gene: initial denaturation at 95°C for 20 seconds, followed by 40 amplification cycles at 95°C for 1 second and 60°C for 20 seconds in each cycle. Relative change in expression of *POLB* was analyzed by using the $2^{-\Delta\Delta C_T}$ method [53]. Results were expressed as the log fold change in gene expression normalized to endogenous reference gene (*GAPDH*) as well

as with the expression of vehicle control at the threshold cycle (Ct). Statistical analysis was performed by means of two-tailed paired t-test with $p < 0.05$ considered as significant.

## Western blotting

Protein was extracted from colonic organoids treated with 100 nmol/L 1,25D or vehicle control (0.1% ethanol) for 6- and 24-hours. Briefly, after treatment, organoids were washed with ice cold PBS and then incubated with RIPA lysis buffer supplemented with protease inhibitor cocktail (Thermo Fisher Scientific, Waltham, MA) for 10 minutes on ice. Lysed organoids were sonicated briefly for 10 seconds, centrifuged at 13,000 rotations per minute for 15 minutes at 4°C and the supernatant was collected. Protein was quantitated using the BCA protein assay kit (Thermo Fisher Scientific, Waltham, MA). Approximately 10–12 μg total protein was used for Western blot assay. The samples were diluted in Laemelli buffer (Bio-Rad, Hercules, CA), boiled for 5 minutes at 95°C. To separate the proteins, samples were subjected to electrophoresis using 4–15% Mini-PROTEAN TGX Stain-Free precast polyacrylamide gels obtained from Bio-Rad, Hercules, CA. Separated proteins were then transferred to a polyvinylidene difluoride membrane. To minimize nonspecific binding, the transferred proteins were blocked using 5% milk in Tris buffered saline with 0.1% Tween 20. The membrane was then probed with primary and secondary antibodies followed by image acquisition on ImageQuant LAS 4000 luminescent imager from (GE Healthcare, Lincoln, NE). Primary antibodies to POLB and GAPDH (catalog numbers ab175197 and 97166S) were obtained from Abcam, Waltham, MA and Cell Signaling Technology Danvers, MA respectively. Secondary antibodies coupled to HRP (catalog numbers 7074S and 7076S) were purchased from Cell Signaling Technology Danvers, MA. Band density was quantitated using ImageJ and normalized to the housekeeping protein to calculate the fold change. Fold change in expression between the vehicle control and 1,25D treated and between the groups was compared using two-tailed paired t-test with $p < 0.05$ considered as significant.

## Luciferase assay

The predicted VDREs in the indel along with a stretch of flanking sequences on their either end was cloned into pGL4.27[luc2P/minP/Hygro] Vector (Promega, Madison, WI) using Infusion Snap Assembly bundle from Takara Bio USA, San Jose, CA. A stretch of 250 bp sequence without the presence of predicted VDRE was also cloned into pGL4.27[luc2P/minP/Hygro] Vector. DNA was isolated from the positive clones containing the inserted sequences using ZymoPure II Plasmid Midiprep kit obtained from Zymo Research Corporation, Irvine, CA, followed by transfection into 293FT cells using Lipofectamine 3000 (Thermo Fisher Scientific, Waltham, MA). Dual-Luciferase(R) Reporter Assay System (Promega, Madison, WI) was used to assess enhancer activity after 24 hours of treatment with 1,25D or vehicle control (0.1% ethanol) according to manufacturer's guidelines. pRL Renilla Luciferase Control Reporter Vector (pRL-TK) was used as an internal control. Relative enhancer activity was determined by dividing the 1,25D-treated values with ethanol-treated values and compared using one-way ANOVA.

Primers used for generating the constructs:

**Construct 1** (insert size 258 bp with predicted VDRE-1):

Forward: 5'-GAGGATATCAAGATCCCTCTGTTTGGGGAATATTCTATAA-3'

Reverse: 5'-CGCCGAGGCCAGATCGCATTTTAATCCACCCTGCT-3'

**Construct 2** (insert size 250 bp, no VDRE):

Forward: 5'-GAGGATATCAAGATCTATTTCCAGTCCTTCTTAGTACTGT-3'

Reverse: 5'-CGCCGAGGCCAGATCCCACCTCCTGCCTGCTCT-3'

**Construct 3** (insert size 250 bp with predicted VDRE-2):

Forward: 5'-GAGGATATCAAGATCAGCCCATTTCTTGCCCGTAG-3'

Reverse: 5'-CGCCGAGGCCAGATCTGAAGCATGGGACTCTTGGACTC-3'

**Construct 4** (insert size 1480 bp with predicted VDREs-1 and 2):

Forward: 5'-GAGGATATCAAGATCCATTTCTTGCCCGTAGCAGTT-3'

Reverse: 5'-CGCCGAGGCCAGATCGCATTTTAATCCACCCTGC-3'

**Chromatin immunoprecipitation (ChIP) assay.** ChIP assay was performed using the SimpleChIP Plus Sonication Chromatin IP Kit from Cell Signaling Technology, Inc. (Danvers, MA) following the manufacturer's instructions. Briefly, organoids were treated with EtOH or 100 nM 1,25D for 6 h, dissociated into single cells by TrypLE Express and cross-linked using 1% methanol free formaldehyde. Cells were then lysed, and the chromatin pellets were fragmented by sonication to an average size of 200- to 1000-bp, using a Fisherbrand Model 120 Sonic Dismembrator (Thermo Fisher Scientific, Waltham, MA). Precleared sonicated extract was diluted into ChIP buffer and subjected to immunoprecipitation by incubating with either a control IgG antibody or 2–4 µg of mouse monoclonal antibody to VDR (Santa Cruz Biotechnology, Dallas, TX) overnight on a rotator at 4 C. The immunoprecipitated DNA fragments were eluted, purified and subjected to PCR using the following pair of primers for predicted VDRE in the indel: forward, 5'-GACAAGAGCAGAAGCAGGAA-3'; reverse, 5'-CTATCAGGCCAAACCCATAAGA-3', which were designed to amplify the indel sequence from coordinates chr8:42193437 to – chr8:42193648 encompassing predicted VDRE and give rise to a 212-bp fragment. A sequence 241 bp encompassing VDRE in the *CYP24A1* promoter was amplified using the primers: forward, 5'-CGAAGCACACCCGGTGAACT-3'; reverse, 5'-CCAATGAGCACGCAGAGGAG-3' from Meyer et al [36] [54] and used as a control. PCR products were resolved on 1% agarose gels and visualized using Sybr safe staining. DNA acquired before precipitation was used to assess the presence of sequence to be amplified following the ChIP procedure and designated "Input."

## Bioinformatic analyses

**Genetic relatedness and ancestry estimates.** Genetic ancestry proportions were estimated with the program ADMIXTURE [55] (v1.3.0) using approximately 255,000 imputed SNPs that were pruned for linkage disequilibrium and filtered for a minor allele frequency greater than 0.05 using PLINK2 [56] (v2.00) commands –indep-pairwise 50kb 1 0.2 and –maf 0.05. The genotype data were then merged with the genotype data from 10 CEU, 10 YRI, and 10 CHB from the 1KGP. ADMIXTURE was run with k = 3 to capture African, European, and possibly Native American or Asian ancestry components of individuals who self-identified as either non-Hispanic White or Black/African American. Principal component analysis (PCA) was performed on the same set of SNPs using PLINK2 with the --pca command. Genetic relatedness was estimated with the same set of SNPs and the PLINK2 command –make-king-table. No pair of individuals was found to have a KING kinship coefficient greater than 0.019.

**Transcriptional responses.** Sequence alignment and gene expression value estimation was performed using the rsem-calculate-expression function of the RSEM [57] v.1.3.1 software package using the STAR [58] aligner. The STAR transcriptome reference was generated from the 1000 genomes Phase2 Reference Genome Sequence (hs37d5) and transcript annotations from the Gencode comprehensive gene annotation GTF (Release 29). *Quality controls:* As QC measures to assess for sample swaps, all mapped RNA-seq samples were checked for both pairwise relatedness and ancestry proportions. The programs angsd [59] (v0.941-17-ge6967e6) and ngsRelate [60] (v2) were used on the resulting alignment files to first estimate genotype likelihoods and then pairwise sample relatedness. Using the angsd-generated genotype likelihoods, ancestry admixture proportions were estimated with the program NGSadmix [61] and the CEU, YRI and CHB data from the fastNGSadmix 1000 Genomes reference panel. An individual's paired RNA-seq samples were

removed from downstream analyses if any of the following were found to be true: (1) the 1,25D and control treatment samples were genetically unrelated; (2) either treatment sample was genetically related to a sample from a different individual; (3) either treatment sample showed an ancestry proportion dissimilar to that estimated from the genotype data. After applying these filters, 53 lines were available for transcriptome profiling (S1 Table).

Genes tested for differential expression were initially filtered for protein coding biotype and expression level by applying a minimum total count threshold of 10 across all samples and then using the filterByExpr function from edgeR [62] (v4.0.16) R package (all R packages were run in R v4.3.1).To account for the paired nature of the data (2 treatment samples per individual) and the inclusion of additional covariates, differential expression was tested with mixed linear models using the dream statistical package [20], which is part of the variancePartition R package [63] (v 1.32.5) and is built on top of the standard limma [64] (v 3.58.1) workflow. In all models, the individual term was treated as a random effect, while covariates and predictor variables were treated as fixed effects. Model covariates included Batch, Age, and Sex ascertained from genotype data.

Single cell analysis of a single organoid line after 24, 48 and 72 hours in differentiation and growth media was performed previously (S1 Methods). The cell populations from these pooled libraries included stem cells, proliferating stem cells, early enterocytes, enterocytes and goblet cells (S1a and S1b Fig). The web-based application CIBERSORTx [65] was used to create a signature matrix from the single cell expression data and impute the relative cell type abundance from the bulk expression data. The total fraction of early enterocytes and enterocytes was included as an additional covariate in the models to control for potential confounding by cell type composition among organoids. Of note, no differences in cell type composition were observed either by treatment or population (S1c Fig).

The following mixed effects *Model 1* was used to test for overall differential expression in response to treatment averaged over both populations:

***Model 1: ~ Individual + Batch + Age + Sex + CellFraction + Population + Treatment***

To test for differences in response to treatment within each population, differences of populations within each treatment and differences of response to treatment between populations, an interaction term whose coefficient represents the difference in response to treatment between the EA and AA populations was added to Model 1 to create *Model 2a*:

***Model 2a: ~ Individual + Batch + Age + Sex + CellFraction + Population + Treatment + Population x Treatment.***

In all cases, the false discovery rate (FDR) was controlled using a Benjamini-Hochberg (BH) adjustment of the estimated p-value. An FDR of 5% or less was considered significant unless otherwise indicated. Because the power for detecting significance is reduced in the second order interaction term in Model 2a, the set of genes tested for this term was restricted to those that showed both DE significance by response to treatment in either EA or AA population and DE significance by population in either treatment condition. To test whether genetic ancestry and self-reported ancestry yielded similar results, the following *Model 2b* was also applied:

***Model 2b: ~ Individual + Batch + Age + Sex + CellFraction + FracAA + Treatment + FracAA x Treatment.***

where FracAA is the proportion of African ancestry as reported by ADMIXTURE.

In box plots, gene expression is reported as the transformed and normalized counts after applying the variance stabilizing transformation (VST) of DESeq2 [66] (v1.44.0). Individual differential gene expression is computed from these VST values and reported as the $\log_2$ fold change of VD/EtOH. As an additional quality check, the expression of the canonical vitamin D responsive gene *CYP24A1* was observed to be significantly upregulated in all 53 lines.

**Gene set enrichment analysis.** Gene set enrichment analysis (GSEA) was performed using the R package SetRank [21]. This method was designed to minimize false positives by taking gene set overlap into account. Briefly, the algorithm inputs a gene set collection from the KEGG, GOBP and REACTOME databases and the list of differentially expressed genes in response to treatment (not filtered by a p-value cut-off). The output includes a setRank value (reflects the importance of the gene set in the gene set network; the higher the value, the more important the gene set), a p-value associated with the

SetRank value (probability of observing a gene set with the same SetRank value in a random network), a corrected p-value (account for overlap with other gene sets) and adjusted p-value (correction of multiple testing). Pathways with the highest SetRank values and associated SetRank p-values<0.05 are shown in the results. The gene set network was created from the SetRank output using the software Cytoscape [67] (v3.10.2). The gene set nodes selected for display were the single significant disease-related gene set (KEGG: Pathways in cancer) and all pSetRank significant nodes.

***Chromatin differential accessibility analysis.*** ATAC-seq read alignment was performed with bwa mem (bwa 0.7.17). Reads were mapped both to the 1000genomes Phase2 Reference Genome Sequence (hs37d5) and the Homo_sapiens_ assembly38.fasta assembly (downloaded from the UCSC Genome Browser website) after the removal of alternate haplotypes. All ATAC results are given for the build 37 assembly. Read filtering was performed with Samtools [68] (v1.10) retaining only uniquely mapped reads with a mapping quality>=10. PCR duplicates were removed with the samtools markup function. *Quality control:* Similar to the RNA-seq samples, ATAC samples were checked for sample relatedness and ancestry proportion using the ngsRelate and fastNGSadmix tools, respectively, and individual's ATAC-seq samples were removed from further downstream analysis if any of the following applied: (1) the individual's 1,25D and control treatment samples were genetically unrelated; (2) either treatment sample was genetically related to a sample from a different individual; or (3) either treatment sample showed an ancestry proportion dissimilar to that estimated from the genotype data.

On the retained samples, ATAC peak calling was performed with MACS2 [69] (2.1.0) using the callpeaks function with the arguments --nomodel, shift = -100, and extsize = 200. The following two criteria then had to be met for individual inclusion in the downstream ATAC DA analysis: (1) MACS peak count was > 30,000 for both treatment samples, and (2) fraction of reads in peaks (FRiP) > 10% for both treatment samples. After applying these filters, ATAC-seq data from 25 organoid lines were included (10AA and 15EA; 15 females and 10 males). In samples mapped to hs37d5, MACS2 called between 30,130 and 111,962 peaks at an FDR less than or equal to 5%, and FRiP scores were between 11% and 36% with an average of 22.4%. Peak differential accessibility analysis was performed with the R package DiffBind [22] (v3.12.0) by employing edgeR (v4.0.16) as the underlying method for the differential peak read count analysis. For the combined population analysis, the DiffBind pipeline was run on a consensus peakset formed from the set of MACS2 peaks present in at least 3 of the 50 samples across both treatments (118,806 peaks). Count data normalization by library size and a standardized differential analysis were performed with edgeR. edgeR normalization factors were computed using the TMM method without precision weights, and then the GLM pipeline was run with tagwise dispersion estimates. As an additional quality check, a significant DA peak with LFC(VD/EtOH)>1 was observed in the *CYP24A1* promoter region of all 25 lines. Differential binding in response to treatment was tested across the 118,806 peakset in both the entire sample set and the separate AA and EA populations using the following *Model 3*:

***Model 3: ~ Individual + Treatment***

A likelihood ratio test was performed for hypothesis testing and FDR controlled using a BH adjustment of the estimated p-value.

Motif enrichment and peak annotation was performed with HOMER [23] (v5.1) using the built-in set of HOMER transcription factor motifs. For HOMER analysis of the set of differentially accessible peaks, the background was chosen to be the set of all 118,806 peaks analyzed for differential accessibility. FDR of 5% or less was considered significant for motif enrichment calls.

**cis-eQTL analyses.** *cis*-eQTL mapping was performed separately on samples from each treatment condition using the program fastQTL [24] (v2.184_gtex). The raw expression data of protein coding genes was first log-transformed and normalized for library size using the vst function of DESeq2. The data was then quantile normalized across all samples from both treatment conditions and, to control for genetic ancestry, the first 3 genetic PCs (see **Genetic relatedness and ancestry estimates**) were regressed out. The sva function from the sva R package

[66] (v3.52.0) found a single significant surrogate variable in the residual expression values, and this was added as a covariate in the fastQTL model. Autosomal bi-allelic variants tested were those having a MAF > 0.10 and falling within a 100 kb window of each tested gene's transcription start site. After applying these filters, there were approximately 5.6 million gene-variant pairs for analysis. To find interaction response-eQTLs, the program mashr [25] (v0.2.79) was run. Approximately 200,000 randomly chosen eQTLs pruned for LD were used to establish the null correlation matrix and the canonical set of covariance matrices alone were used for fitting the mashr model. Following the recommendation in Urbut et al [25], an eQTL was considered a response-eQTL if 1) its lfsr<0.05 in either treatment condition and 2) either the effect size posterior means had the same sign in the two conditions and differed by a factor > 2 or had opposite signs. The significant results from mashr were overlapped with the roughly 3.5 million eQTLs from sigmoid and transverse colon tissue listed in the 48 cross-tissue metasoft analysis of GTEx v7 with metasoft posterior probability (m-value) >0.90 in either tissue. We find that 53.0% of listed GTEx eQTLs have m > 0.90 in either sigmoid or transverse colon tissue; when conditioning on these eQTLs with m > 0.90 in at least one of the two tissues, 60.9% have m > 0.90 in both tissues. For our tested eQTLs overlapping with those in the GTEx metasoft database, m > 0.90 in either colon tissue for 78.6% of 31,533 eQTLs with lfsr<0.05 in both vehicle control and 1,25D; for 88.5% of the 295 eQTLs with lfsr<0.05 only in vehicle control; and for 37.7% of the 875 eQTLs with lfsr<0.05 only in 1,25D.

We intersected the positions of the discovered reQTLs with the set of 400 bp regions centered on the summits of the DA peaks. For the peaks with overlapping reQTLs, using the genotypes and LFCs from the resultant set of reQTLs/ eGenes/DARs, we performed mediation analysis with the R package bmediatR [70] (v0.1.3) in order to identify whether these reQTLs might impact differential expression through alterations in chromatin accessibility. In the bmediatR framework, DARs act as a mediator in the complete and partial mediation models. The posterior probabilities for 4 mediation models (complete, partial, col-localization, and non-mediation) were then plotted using the bmediatR plot_posterior_bar function.

To explore potential functional relevance of these interaction reQTLs for human traits and diseases, we assessed overlap of these variants (or those in strong linkage disequilibrium) with variants from several sources: 1) publicly available databases (e.g., UK Biobank and GWAS catalog) and 2) previously published studies.

**Chromatin differential accessibility QTL analyses.** cis-daQTL mapping was performed on the LFC of significant differentially accessible ATAC peaks for the 25 individuals with passing ATAC QC metrics, using all imputed bi-allelic variants falling within 100 kb of peak start or end positions. Peak LFC was calculated from the DESeq2 VST transformation of peak read counts, and association mapping was performed using the program MatrixEQTL (v2.3), applying both linear and ANOVA models to detect additive and dominant associations, respectively. For the linear model, of 2,110,074 DA peak-variant tests, 132,578 tests had nominal p-value<0.05 with 594 having an FDR<0.05; for the ANOVA model, of 2,110,074 tests, 106,277 tests had nominal p-value<0.05 with 2,108 having an FDR<0.05.

**POLB selection signal analyses.** iHS values reported for rs2272733 in 1 KG Phase 3 CEU and YRI populations based on the approach from Johnson KE et al [71]. applying their updated normalization method; associated empirical p-values are estimated from the normal distribution function. Weir Fst, cross-population extended haplotype homozygosity (XP-EHH) and Tajima's D statistics are from Pybus M et al [72]; associated p-values are calculated from genome-wide rank scores. The visual haplotype was created from 1KGP phased genotype data for CEU and YRI.

**Inference of POLB indel genotype.** The genotype of the indel (nssv16196380) proximal to the POLB promoter is imputed in our samples using the 1KGP Phase 3 dataset as a reference. Overall, the posterior probability for indel genotype imputation was > 0.942. For the lines used for validation, the posterior probability for the indel was > 0.999. We also more directly infer indel genotype from the genotyped SNP rs2272733 because the two are in perfect LD ($r^2 = 1$) in the 1 KG CEU and YRI samples, as well as in our samples. We also validated genotype status of samples inferred to be homozygous deletion from ATAC-seq coverage of the region.

## Supporting information

**S1 Fig. Cell composition from a previous single cell sequencing dataset.** As described in the Methods, organoids from a single individual were cultured in growth and differentiation media for 24, 48 and 72 hours and single cell RNA-sequencing was performed. Data was utilized to assess cell composition measured as the fraction of early enterocytes and enterocytes for downstream analyses in this study. a) **Uniform Manifold Approximation and Projection** (**UMAP**) of single cell sequencing. The UMAP plot shows 7 clusters of cell types. b) **Cell type markers.** Clusters were annotated based on expression of cell type marker genes previously reported in the literature (see references for genes in Methods). c) **Percentage of early enterocytes and enterocytes by treatment and population**. The percentage of early entero-cytes and enterocytes was used to control for cell composition in downstream analyses. There were no differences by treatment or population.
(TIFF)

**S2 Fig. Genomic responses to 1,25 vitamin D (VD) treatment. a) Principal component (PC) plot of transcriptional responses.** VD transcriptional responses measured by RNA-seq were separated by PC2, which accounted for 11% of the variance. b) **PC plot of chromatin accessibility responses**. VD chromatin accessibility responses measured by ATAC-seq were separated by PC5, which accounted for 4% of the variance. Separation along PC1 represents variation due sex-linked chromatin accessibility. c) **SetRank network plot.** To visualize interactions of top enriched pathways of DE genes, we used the network output of SetRank plotted using the program Cytoscape (v3.10.2). We show the inter-actions between the only disease-associated enriched pathway called "pathways in cancer" (KEGG hsa05200) with the top enriched pathways. The node fill color reflects the SetRank corrected p-values with blue to red indicating decreasing p-values. The edge arrows represent interaction from least significant gene set to more significant gene set.
(TIFF)

**S3 Fig. Differentially accessible (DA) peak analysis. a) DA peak enrichment.** For DA peaks (FDR < 5%), a broader distribution of distances to the transcription start site (TSS) of the nearest gene was observed relative to peaks that were not DA (FDR > 99%). This observation is in line with the finding that DA peaks were significantly depleted in promoter regions (3.6% vs. 12.4%, respectively; hypergeometric test $p = 1.54 \times 10^{-89}$) compared to intronic or intergenic regions. b) **Correlation of DE genes and DA peaks in promoter regions**. A total of 85 tested protein-coding genes were found to have DA peaks falling in their promoters. Of these, 73/85 (86%) were DE, a 1.3-fold enrichment over genes without DA peaks in their promoters (hypergeometric test $p = 8.18 \times 10^{-5}$). The corresponding DE and DA effect sizes for these 73 genes were strongly correlated ($r = 0.85$; $p < 2.2 \times 10^{-16}$). c) **Vitamin D response element (VDRE) peak enrichment**. The VDRE motif was found within 250 bp of a DA peak center for 75% of DA peaks with large effects (i.e., LFC > 1), 46% with moderate effects (i.e., 0 < LFC < 1), and only 3.8% of DA peaks with reduced effects (i.e., LFC < 0). These patterns were similar to VDR enrichment patterns.
(TIFF)

**S4 Fig. 1,25 vitamin D (VD) genomic responses by ancestry. a-b) Genetic ancestry. a)** Principal component PC1 vs PC2 of genotype SNP data with self-identified White in blue dots and self-identified African-American in red dots, and **b)** PC1 vs fraction of African (i.e., YRI) ancestry with self-identified White in blue dots and self-identified African-American in red dots. Genetic ancestry proportions were estimated with the program ADMIXTURE (v1.3.0) using approximately 255,000 imputed SNPs. c) **Correlation of transcriptional response effect sizes in AA and EA lines (Model 2a)**. There was significant correlation between effect sizes (i.e., LFC) for DE genes with treatment between AA and EA ($r = 0.960$; $p < 2.2 \times 10^{-16}$). This result was similar to results for z-scores shown in Fig 1e. d-g) **Correlation of treatment effects between models that include fraction African ancestry (Model 2b) and self-identified race (Model 2a).** We assessed treatment responses using both fraction African ancestry and self-identified race. Comparison of both effect sizes and z-scores for the treatment and interactions terms of the models showed very high correlations and similar power. For

the interaction term, there was near perfect correlation between results from the two models (r > 0.99; p < 2.2 x 10$^{-16}$). h) **Genome-wide transcriptional responses by ancestry.** To determine whether there were differences in overall absolute response to 1,25D treatment, we compared the between population transcriptional effect size magnitude difference ($|LFC_{AA}|$ - $|LFC_{EA}|$) to the expected null distribution generated from permuted samples. Using this approach, neither population showed a stronger overall genome-wide response to treatment (p = 0.28).
(TIFF)

**S5 Fig. eQTLs. a-b) mashr results for eQTLs significant (lfsr<0.05) in at least one treatment condition. a)** mashr local false sign rate (lfsr) for variants in 1,25D (y-axis) versus control (x-axis) showing rs2272733-*POLB* (red dot). Dashed line represents y = x. **b)** mashr effect size posterior mean (pm) for variants in VD (y-axis) versus control (x-axis) showing rs2272733-*POLB* eQTL (red dot). Dashed line represents line y = x. **c-d) *ZC3HAV1* eQTLs. c)** MatrixEQTL adjusted p-values were plotted as a function of physical position for variants within a 2 Mb window centered on *ZC3HAV1* for VD (red dots) or control (blue dots) treatment conditions. Vertical lines represent position of gene. **d)** *ZC3HAV1* shows association with rs12672468 genotype only in 1,25D treatment condition.
(TIFF)

**S6 Fig. bmediatR results for 7 reQTLs overlapping DA regions.** bmediatR assessed 4 models of reQTL mediation by a DA peak and showed a high posterior probability for complete or partial mediation only for the indel-*POLB* reQTL. **a)** 8:42190387:A:<CN0>:42194352-PK105786-*POLB.* **b)** rs4242349-PK108142-*FER1L6.* **c)** rs28411912-PK82292-*SGMS2.* **d)** rs28585029-PK82293-*SGMS2.* **e)** rs28626074-PK82293-*SGMS2.* **f)** rs2112161-PK86623-*ARHGEF28.* **g)** rs4703608-PK86628-*ARHGEF28*
(TIFF)

**S7 Fig. Core haplotypes in *POLB* region. a & b) Core haplotype in *POLB* region in YRI and CEU populations.** Of the 33 SNPs that were daQTLs and *POLB* reQTLs, 10 SNPs (rs2272733, rs7002979, rs10958714, rs13278231, rs13270698, rs7462320, rs7463029, rs3136717, rs75422254, rs11990332) were found to define a core haplotype where pairwise SNP $r^2$ > 0.90 in both 4a) YRI and 4b) CEU 1KGP populations. c) Core haplotype spans *POLB* and *IKBKB.* The core haplotype spans the genes *POLB* and *IKBKB.* The tagging SNP rs2272733 showed no association with *IKBKB* expression nor did *IKBKB* show a differential response to 1,25 vitamin D treatment, while *POLB* showed an association with genotype only in response to 1,25 vitamin D treatment. Expression shown as the DESeq2 variance stabilized transformation (vst) of normalized read counts.
(TIFF)

**S8 Fig. Tissue-specific *cis*-eQTL effects of rs2272733 on *POLB* expression from the Adult Genotype Tissue Expression (GTEx) project. a) Effect sizes of rs2272733 in *POLB* expression.** The effects of rs2272733 on *POLB* expression differ by tissue type. Shown here are GTEx tracks for different tissues showing effect sizes by color. rs2272733 is shown as the vertical line. **b) rs2272733 *cis*-eQTL effects on *POLB* in the same direction as 1,25 vitamin D colonic responses**. Tissues that showed similar direction of *POLB* cis-eQTL effects for rs2272733 included skin (sun exposed and non-sun exposed), esophagus and, to a lesser extent, testis. **c) rs2272733 *cis*-eQTL effects on *POLB* in the opposite direction as 1,25 vitamin D colonic responses**. Tissues that showed opposite direction of *POLB* cis-eQTL effects for rs2272733 included skeletal muscle, nerve, lung and subcutaneous adipose tissue.
(TIFF)

**S1 Table. Participant information, Ancestry estimates and Quality Controls for RNA- and ATA-seq datasets.**
(XLSX)

**S2 Table. Differential expression by treatment and population.**
(XLSX)

**S3 Table. Gene set enrichment analysis.**
(XLSX)

**S4 Table. Differential accessibility by treatment and population.**
(XLSX)

**S5 Table. Differential accessibility peak annotation.**
(XLSX)

**S6 Table. Transcription factor enrichment.**
(XLSX)

**S7 Table. Response-expression quantitative trait loci (reQTL) mapping.**
(XLSX)

**S8 Table. Differential accessibility quantitative trait loci (daQTL) mapping.**
(XLSX)

**S9 Table. Ancient DNA database results for rs72733.**
(XLSX)

**S1 Methods. Methods describing single cell dataset for a single organoid line cultured in differential and growth media for 24, 48 and 72 hours.**
(DOCX)

## Acknowledgments

We thank Dr. Anna Di Rienzo and Dr. Luis Barreiro for helpful discussion. We thank Dr. Candace Cham and Dr. Eugene Chang for their assistance with organoid culturing. We thank study participants.

## Author contributions

**Conceptualization:** David Witonsky, Sonia Kupfer.

**Data curation:** David Witonsky.

**Formal analysis:** David Witonsky, Bharathi Laxman, Sonia Kupfer.

**Funding acquisition:** Sonia Kupfer.

**Investigation:** Bharathi Laxman, Margaret C. Bielski, Kristi M. Lawrence.

**Methodology:** David Witonsky, Bharathi Laxman, Hina Usman, Margaret C. Bielski.

**Project administration:** Sonia Kupfer.

**Supervision:** Sonia Kupfer.

**Writing – original draft:** David Witonsky, Bharathi Laxman, Sonia Kupfer.

**Writing – review & editing:** David Witonsky, Bharathi Laxman, Sonia Kupfer.

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
