## [Decision Letter · Decision Letter 0]

7 Sep 2025

PGENETICS-D-25-00725

Genomic profiling of active vitamin D colonic responses in African- and European-Americans identifies an ancestry-related regulatory variant of POLB

PLOS Genetics

Dear Dr. Kupfer,

Thank you for submitting your manuscript to PLOS Genetics. After careful consideration, we feel that it has merit but does not fully meet PLOS Genetics's publication criteria as it currently stands. Therefore, we invite you to submit a revised version of the manuscript that addresses the points raised during the review process.

Please submit your revised manuscript within 60 days Nov 06 2025 11:59PM. If you will need more time than this to complete your revisions, please reply to this message or contact the journal office at plosgenetics@plos.org. Please include the following items when submitting your revised manuscript:

We look forward to receiving your revised manuscript.

Kind regards,

Heather J Cordell

Academic Editor

PLOS Genetics

Hua Tang

Section Editor

PLOS Genetics

Aimée Dudley

Editor-in-Chief

PLOS Genetics

Anne Goriely

Editor-in-Chief

PLOS Genetics

**Additional Editor Comments:**

Reviewer #1:

Reviewer #2:

**Journal Requirements:**

At this stage, the following Authors/Authors require contributions: Bharathi Laxman, Hina Usman, Margaret Bielski, Kristi Lawrence, and Sonia Kupfer. Please ensure that the full contributions of each author are acknowledged in the "Add/Edit/Remove Authors" section of our submission form.

The list of CRediT author contributions may be found here: https://journals.plos.org/plosgenetics/s/authorship#loc-author-contributions

https://journals.plos.org/plosgenetics/s/submission-guidelines#loc-parts-of-a-submission

- ® on pages: 21, 23, 24, and 25

- TM on pages: 23, and 24.

5) We notice that your supplementary Figures are included in the manuscript file. Please remove them and upload them with the file type 'Supporting Information'. Please ensure that each Supporting Information file has a legend listed in the manuscript after the references list.

Potential Copyright Issues:

i) Figure 3F. Please (a) provide a direct link to the base layer of the map (i.e., the country or region border shape) and ensure this is also included in the figure legend; and (b) provide a link to the terms of use / license information for the base layer image or shapefile. We cannot publish proprietary or copyrighted maps (e.g. Google Maps, Mapquest) and the terms of use for your map base layer must be compatible with our CC BY 4.0 license.

ii) We note that Figure 1A is created through BioRender. Please confirm that you hold a Premium account and provide a pdf copy of the CC BY 4.0 Licence as provided by BioRender. For instructions on how to generate a CC BY 4.0 license for your figure, please see the guidelines here: https://help.biorender.com/hc/en-gb/articles/21282341238045-Publishing-in-open-access-resources.

If you are using the free assets from BioRender, we are unable to publish these images as they are licenced under a stricter licence than CC BY 4.0. In this case we ask you to remove the BioRender images and replace them with open source alternatives.

See these open source resources you may use to replace images / clip-art:

- https://bioart.niaid.nih.gov/

- https://bioicons.com/

- https://healthicons.org/

- https://scidraw.io/

- https://reactome.org/icon-lib

- https://www.phylopic.org/images

- https://journals.plos.org/plosbiology/article?id=10.1371/journal.pbio.3002395

**Reviewers' comments:**

Reviewer's Responses to Questions

**Comments to the Authors:**

Reviewer #1: This is an interesting and thorough study of the effect of 1,25D on color organoids, with emphasis on the POLB findings. In particular, this reviewer appreciated the multi-omic approach and thoughtful consideration of ancestry. Below are concerns about the overall approach, as well as some comments related to interpretation.

1) I am a bit confused by the information presented in Figures 1D and 1F the 1D, the authors state ~4000 peaks were significantly associated with 1,25D treatment. But in 1F, the number by ancestry (one alone or both) is only ~500, as the legend states.

2) For eQTL analysis, the author states that they used BRIdGE and MatrixQTL to identify eQTLs; however, there is no mention of multiple testing correction commonly done in eQTL analysis. An FDR (Or posterior probability cut-off) is not sufficient for the number of SNPs tested, of the number of genes assessed. More up-to-date methods have now been created to assess Context-specific eQTLs (Meta-Tissue, MashR).

3) In the eQTL analysis, no covariate (age, sex, etc) seems to have been added to the analysis. Were both ancestors mapped together? This may pose a real issue if no PC were added. No SVA or PEER was included to correct for hidden variables in gene expression.

4) The authors focus on POLB because of the difference in transcriptional response with treatment. In the LFC on ATAC seq peak near this locus, only 9 to 10 points are shown. Why are there so few samples in this analysis? The methods are stated 12-13 in each group.

5) 3 models of DE analysis are described in the methods, but only one is shown in the results (Model 1). The other. Models are very interesting, and the results of those should also be presented.

6) It would be interesting to know if the eQTL identified after treatment are located in DA genomics regions, with the idea that some SNPs may not exert a regulatory effect until they are located within open chromatin.

7) Missing from the discussion is a clear delineation of limitations, such as the use of organoid cultures, which may differ from colonic tissue in vivo, the relatively small sample size, and the power to detect association inherent in the study, especially once divided by ancestry.

8) People of African Ancestry are known to have lower levels of Vitamin D as well as higher levels of VDR. How would these differences relate to your findings? While genetic regulation is interesting, would the long-noted difference long noted blunt the findings?

Minor Comments

1) As a result, the authors state that they controlled for Ancestry (line 102) in the DE analysis (Model 1). I believe they mean population as Ancestry is controlled for in Model 3.

Reviewer #2: review will be uploaded as attachment

**Have all data underlying the figures and results presented in the manuscript been provided?**

Reviewer #1: None

Reviewer #2: Yes

PLOS authors have the option to publish the peer review history of their article (what does this mean? ). If published, this will include your full peer review and any attached files.

**Do you want your identity to be public for this peer review?** For information about this choice, including consent withdrawal, please see our Privacy Policy .

Reviewer #1: No

Reviewer #2: **Yes:** Syed Munim Husain

**Figure resubmission:**
---

## [Decision Letter · Decision Letter 1]

5 Dec 2025

Dear Dr Kupfer,

We are pleased to inform you that your manuscript entitled "Genomic profiling of active vitamin D colonic responses in African- and European-Americans identifies an ancestry-related regulatory variant of POLB" has been editorially accepted for publication in PLOS Genetics. Congratulations!

Yours sincerely,

Heather J Cordell

Academic Editor

PLOS Genetics

Hua Tang

Section Editor

PLOS Genetics

Aimée Dudley

Editor-in-Chief

PLOS Genetics

Anne Goriely

Editor-in-Chief

PLOS Genetics

BlueSky: @plos.bsky.social

Comments from the reviewers (if applicable):

Reviewer's Responses to Questions

**Comments to the Authors:**

Reviewer #1: The authors have addressed all comments fully.

Reviewer #2: The authors have fully addressed my previous comments. Well done on the substantial and excellent work.

**Have all data underlying the figures and results presented in the manuscript been provided?**

Reviewer #1: Yes

Reviewer #2: Yes

PLOS authors have the option to publish the peer review history of their article (what does this mean? ). If published, this will include your full peer review and any attached files.

**Do you want your identity to be public for this peer review?** For information about this choice, including consent withdrawal, please see our Privacy Policy .

Reviewer #1: No

Reviewer #2: **Yes:** Syed Munim Husain

**Data Deposition**

http://datadryad.org/submit?journalID=pgenetics&manu=PGENETICS-D-25-00725R1

**Press Queries**

---

## [Editor Report · Acceptance letter]

PGENETICS-D-25-00725R1

Genomic profiling of active vitamin D colonic responses in African- and European-Americans identifies an ancestry-related regulatory variant of POLB

Dear Dr Kupfer,

We are pleased to inform you that your manuscript entitled "Genomic profiling of active vitamin D colonic responses in African- and European-Americans identifies an ancestry-related regulatory variant of POLB" has been formally accepted for publication in PLOS Genetics! Your manuscript is now with our production department and you will be notified of the publication date in due course.

With kind regards,

Anita Estes

PLOS Genetics

On behalf of:
